# Numerical Investigation of Parameters Influencing Back-Thrust Development in Outer Wedge Fronts of Fold-and-Thrust-Belt Systems

Saeed Mahmoodpour[1*], Florian Duschl[1], Michael C. Drews[1]

[1]Geothermal Technologies, Technical University of Munich, Munich, 80333, Germany

[*]Correspondence to Saeed Mahmoodpour (saeed.mahmoodpour@tum.de)

**Key points**

- Material strength is recognized as the most important factor for back-thrusting
- Increasing the friction coefficient of the décollement decreases likelihood of back-thrusting
- Strength contrast between taper and décollement is the main driver of back-thrusting and compares well with field observations

**Abstract**

Thrusting in fold-and-thrust belts can manifest in different styles. Here we investigate the parameters influencing back-thrust development over fore-thrust development at the frontal part of fold-and-thrust belts using numerical
geomechanical forward modelling. We vary the strength of the material involved, dip and friction of the décollement and displacing boundary conditions to examine the impact of these properties on back-thrust development. The results of this numerical sensitivity analysis reveal that back-thrusting mainly increases with increasing material strength and decreasing friction coefficient of the décollement. Décollement dip has a less prominent impact on back-thrusting, but decreasing the décollement's dip angle enhances back-thrusting
likelihood. In summary, we find that the contrast between the work necessary to shear the wedge material and the work necessary to slide along the décollement is the main driver for initiating back-thrusting (high contrast) over fore-thrusting (low or even negative contrast), which compares well with field observations. In addition, we also investigate and discuss the effect of a pure lateral displacement rate boundary condition vs. a coupled lateral and along-décollement displacement rate boundary condition on numerical simulations of back-thrusting
development.

**Keywords:** Geomechanics; Fold and Thrust Belt; Back-thrust; Sensitivity analysis

## 1 Introduction

A fold-and-thrust-belt is a geological feature which forms due to compression and is characterized by folded and
thrusted rock layers. Resulting thrusts can be divided into fore-thrusts and back-thrusts based on their development directions relative to the layers' displacement. In this study, back-thrust is assumed as a vergence towards the rear of the wedge with a succession of faults that chronologically develop towards the back wall (displacing wall). These structures are observed in major mountain ranges across the world such as the European Alps (Brisson et

al., 2023), the Himalayas (Mugnier et al., 2022), and the Rocky Mountains (McMechan, 2023), but also in sub-sea accretionary prisms and at frontal parts of deltaic and mass transport deposits (Bilotti & Shaw, 2005). Typically, the development of back-thrusts occurs between the axis of the main orogenic uplift zone and its foredeep; here, convergence is accommodated by thrust faults and folding. Analysing the progressive development of fold-and-thrust-belt systems provides essential insights into mountain-building processes, geological history and the mechanisms which are shaping the Earth's crust in tectonic convergence zones.

Hubbert and Rubey (1959) were of the pioneers who examined the mechanisms behind the development of fold-and-thrust-belt systems and built the basis for the development of the critical taper theory (Eq. 1 for the dry sand wedge), as formulated by Dahlen (1990). The critical taper theory examines the force balances to form a wedge (taper) through compression and provides an analytical tool which considers the relationship between pore pressure, stress, and the geometry of fold-and-thrust-belts. Based on this approach, material strength and basal friction of a wedge are used to characterize critical taper development. Assuming the predefined material strength and basal friction in a wedge with known basal dip angle, minimum and maximum values are calculated as the critical taper angles (difference between surface slope and basal dip angle, $\alpha + \beta$ in Eq. 1). When the taper angle is equal to $\alpha + \beta$, the wedge is assumed to be stable. When the taper angle is lower than the minimum value, the surface dip angle increases through thrusting resulting in higher wedge thickness. This process continues until the taper angle reaches the minimum value, i.e. critical state. Accretion of new material at the toe of the wedge then keeps the surface angle stable. In cases where the taper angle exceeds the maximum value, the surface angle decreases through normal faulting at the wedge front (Ruh et al. 2012):

$$\alpha + \beta \approx (\frac{1-sin\varphi}{1+sin\varphi})(\beta + \mu_b) \qquad\qquad\qquad \text{Eq. 1}$$

Where $\alpha$, $\beta$, $\varphi$, and $\mu_b$ are the surface dip angle, basal dip angle, angle of internal friction, and basal friction coefficient, respectively. The governing equations of the involved processes in the evolution of fold-and-thrust-belt systems and their interactions are highly non-linear. Also, the boundary and displacement conditions are not the same for all cases. These are challenges for the unified analytical approaches, such as the critical taper theory. Considering the critical taper theory in a system with predefined material properties (constant $\varphi$ in Eq. 1), increasing basal friction coefficient (stronger décollement) results in a larger taper angle. This system will try to reach a stable condition through fore-thrust development at the front of the wedge. In contrast, decreasing the basal friction coefficient (weaker décollement) results in a smaller taper angle. In this system, back-thrusting at the rear of the wedge tries to increase the taper angle. These relationships have been extensively explored through numerical and analogous sandbox experiments. Numerous factors, including the surface features (Marques and Cobbold, 2006), thickness and strength of the overburden layers (Lohrmann et al., 2003; Teixell and Koyi, 2003), material deposition or erosion (Cruz et al., 2010; Smit et al., 2010), properties of the basal décollement (Koyi and Vendeville, 2003; Smith et al., 2003), the presence of weak layers within the rock sequence (Ruh et al. 2012), and as well as the difference in material strength of the top layers to basal décollement (Couzens-Schultz et al., 2003), have been identified as influencing factors for the development of back-thrusts (Zhou et al., 2016).

Seely (1977) explained the presence of the low basal friction through overpressure development and presence of weak material in this zone. The idea of back-thrusting in a system with a weak décollement is confirmed through experimental investigations (Gutscher et al., 2001; Couzens-Schultz et al., 2003; Bonini, 2007). Likewise, (Bilotti

& Shaw, 2005) attributed back-thrusting in the Niger Delta to the low basal friction which results from overpressure in this zone. Furthermore, the dip angle of the basal décollement can play a role in back-thrust development. MacKay (1995) examined this parameter and claimed that fore-thrusts develop in a system with a low basal angle. A system with higher basal angle will therefore favor back-thrusting. Albertz & Sanz (2012) used numerical simulation to examine fold-and-thrust-belt systems and found that systems with over-consolidated material involved (high strength materials) resulted in back-thrusting. Their findings suggested that rigid materials exhibit locally concentrated strain zones, whereas soft materials display diffusive plastic strain behaviour. Similarly, Cubas et al. (2016), by using numerical simulation, mentioned that there is a chance of back-thrusting in systems with high pre-consolidated material, high basal dip angle, and low basal friction coefficient. Further numerical investigations by (Del Castello and Cooke, 2007) showed that development of back-thrust requires more energy than fore-thrust formation and overpressure development in the basal layer enhances the likelihood of back-thrusting.

Nevertheless, the conditions that lead to back-thrusting are not thoroughly examined in detail. Recently, van Hagke et al. (2023) investigated the friction coefficient and dip angle of the décollement in Columb wedges (a friction wedge used to convert forces) and observed that back-thrust-dominated wedges tend to develop in configurations with very low basal dip angle (less than 0.5 degrees) and a low friction coefficient. Increasing the friction coefficient or dip angle of the décollement causes the development of pop-up structures and finally fore-thrusting may be the dominate deformation style. However, the examination of the real field cases also suggests that friction coefficient and basal décollement angle might not be the only reason behind back-thrusting highlighting the importance of a more systematic examination of back-thrust development, which addresses the different controlling parameters.

In this context, employing a large strain geomechanical forward modelling scheme combined with dynamic elasto-plastic behavior of the involved materials can offer valuable insights. The elasto-plastic behavior of the soil can be tracked through the critical state soil mechanics model (Roscoe et al., 1958; Wood, 1990). Models derived from critical state theory have proven highly successful in describing mechanical behavior of geological formations such as subduction zones and offshore accretionary prisms systems (Song et al. 2011; Flemings and Saffer 2018), often achieving this through a finite set of inputs (Albertz & Sanz 2012; Albertz & Lingrey 2012). Thereby, critical state soil mechanics models make a relationship between the porosity changes and variation in the deviatoric and mean effective stresses through the numerical simulation of the subsurface deformation (Crook et al., 2006). Incorporation of critical state soil mechanics into numerical geomechanical forward modelling can help understanding the evolution of complex geological features and has been successfully employed in the past to study deformation in subduction zones, accretionary prisms (Obradors-Prats et al., 2017; Gao et al., 2018; Nikolinakou et al., 2023) and other compressive geological features (Heidari et al., 2020; Obradors-Prats et al., 2023). Limit analysis theory has also been applied in the literature to study fold-and-thrust belt systems. In this regard, Mary et al. (2013) demonstrated that the location of faults and their lifetimes are governed by deterministic chaos. Robert et al. (2019) used this approach to investigate the impact of syn-tectonic sedimentation on stresses in a ramp propagation fold. These stress values are essential for examining fracture development, their orientation, and the resulting fluid flow patterns in basin analysis. Adwan et al. (2024) applied the limit analysis approach to study stress distribution at the lateral termination of a thrust fold system. The fast run time of simulations using

this approach enabled the authors to conduct a high number of simulations to analyse the effects of basement and fault friction angles on the failure pattern.

In the presented study, we narrow our focus to the limited controlling parameters of fore-thrust and back-thrust development at the frontal part of fold-and-thrust belts by performing a parametric study by the means of numerical geomechanical forward modelling. Thereby, we assume that a taper has already formed and is advancing over a total timespan of 7 Ma. Our study also considers sedimentation while deformation by modelling a single depositional event. We use a constitutive law, which resembles critical state soil mechanics (Soft Rock

SR3, Crook et al., 2006) and vary pre-consolidation pressure as proxy for material strength, initial taper dips, friction coefficient of the décollement, and displacement boundary conditions to investigate their impact on the resulting shape and orientation of evolving thrusts. A sensitivity analysis is performed on these parameters to assess the influence of them on back-thrust development and the results are discussed in the context of previous studies and real-world examples.

**2 Methodology**

In this study, a plane-strain condition is employed to model geomechanical processes through the Elfen software (Rockfield, 2017). To minimize potential boundary effects from additional sediment loading, a quasistatic criterion is considered during the modelling. Explicit and Lagrangian methods are utilized to discretise the equations, taking into account the finite strain nature of the system.

**2.1 Meshing**

Adaptive remeshing is employed to track the large-strain behaviour efficiently within reasonable computational time. After each calculation step, element distortion in each area is compared with a predefined limit. If the distortion area error exceeds a critical value (here 20), remeshing is initiated, with the mesh size being a non-linear function of plastic strain through an interpolation function:

- Mesh size = 400m for Plastic strain = 0
- Mesh size = 400m for Plastic strain = 0.1
- Mesh size = 300m for Plastic strain = 0.2
- Mesh size = 200m for Plastic strain > 0.5

This adaptive remeshing prevents premature termination of the simulation and offers benefits such as reducing

calculation errors, tracking deformation effectively, and accelerating calculations in low-affected areas with larger-sized elements.

**2.2 Model setup and initiation**

**2.2.1 Geometry**

The initial geometry presents the very frontal part of a fold-and-thrust belt in a generalized two-dimensional plane-

145 strain domain (Figure 1A). In this geometry, six layers (layer 5 at the bottom and layer 1 being the top most layer, while layer 0 is deposited during the simulation) are distinguished and characterized by homogeneous and isotropic material properties. The surface angle α of the initial geometry (layer 1) and the angle of the décollement

β are varied in this study. The length of the system is 50 km and layer thicknesses is $350m$ for layer 5, $500m$ for layer 4, $400m$ for layer 3, $1000m$ for layer 2, and $500m + 50km \times (tan(|\beta|) + tan(|\alpha|))$ for layer 1 based on the Figure 1. The top surface of layer 0 is flat and located $11km$ above the lower boundary of the model. The décollement is defined by a frictional surface between the upper and lower sections and is placed between layer 2 and 3. The décollement is divided into areas of different friction by a discontinuity point, as illustrated in Figure 1 (marked with X). The friction coefficient on the right side of the discontinuity point is varied in this study, while the left side is restricted from displacement (i.e. layer 2 = layer 3).

**2.2.2 Displacement conditions**

A predefined movement boundary condition is assigned to the right boundary of the system (Figure 1A). In the geomechanical analysis, the only criterion for the displacement rate is the quasistatic condition. This means that the loading is implemented so gently that at each given time step the deformation happens in a static condition and the inertia impacts are negligible. As long as this condition is not violated, the final result remains independent of the displacement rate. The right-hand section of the décollement is vertically constrained to avoid upward motion during the leftward displacement of the upper layers, ensuring no uplift due to vertical relaxation when the overburden load decreases to zero. This assumption causes high plastic strain development in this area during the initialization period, which disappears after horizontal shortening is initiated. Horizontal shortening is modelled with an equal rate displacement rate of layers 2, 1 and 0, whereas layer 0 is deposited after a prescribed time. The left boundary is restricted from horizontal movement. The bottom boundary remains restricted in all directions throughout the simulation (Figure 1A). In addition, we also test an additional boundary condition where movement of the overburden layers along the décollement is fixed to the same rate as the movement of the right-hand side boundary (Figure 1A). This boundary condition has been employed in other experimental and numerical studies (Ruh et al. 2012; Obradors-Prats et al. 2017; v Hagke et al. 2023) and may lead to differences in the style of deformation (Vendeville 1991, 2007; Zhou et al. 2016). We implement the additional boundary condition by applying a constant displacement rate to the bottom surface of layer 2, rightward of the discontinuity point (Figure 1A).

**2.2.3 Model initiation and key modelling steps**

Before horizontal shortening is initiated from the right-hand side of the model, the initial 0.5 million years (Ma) of the simulation are dedicated to allow for gravitational settling of the system (Figure 1B). Thereby, the total initial height of the model will decrease due to gravitational compaction with horizontal stress being allowed to build up following an effective stress ratio $K_0 = \frac{horizontal\ effective\ stress}{vertical\ effective\ stress} = 0.8$. Subsequently, horizontal displacement starts with two different movement rates: 300 m from 0.5 Ma to 1 Ma, followed by 6500 m from 1 Ma to 7 Ma (Figure 1C). Initially, only pure horizontal displacement occurs, representing tectonic convergence. Starting from 5 Ma onward, horizontal displacement is coupled with simultaneous deposition and increased vertical loading to account for additional sedimentary deposition and its impact on deformation evolution (Figure 1D). Additionally, the selection of time steps is crucial for successful simulation runs. While a large time step helps to reduce the simulation runtime, it may endanger the quasistatic state of the simulation. Therefore, through trial and error, the time steps are chosen to maintain the quasistatic state of the solution while being large enough

to improve simulation speed. Although, as shown in Figure 1, the décollement is assumed to be a straight line in the initial geometry. While real field cases may exhibit some degree of elastic curvature. This curvature can influence stress orientation and deformation style (Willett and Schlunegger, 2010). However, for simplicity, it is assumed to be a straight line in this study.

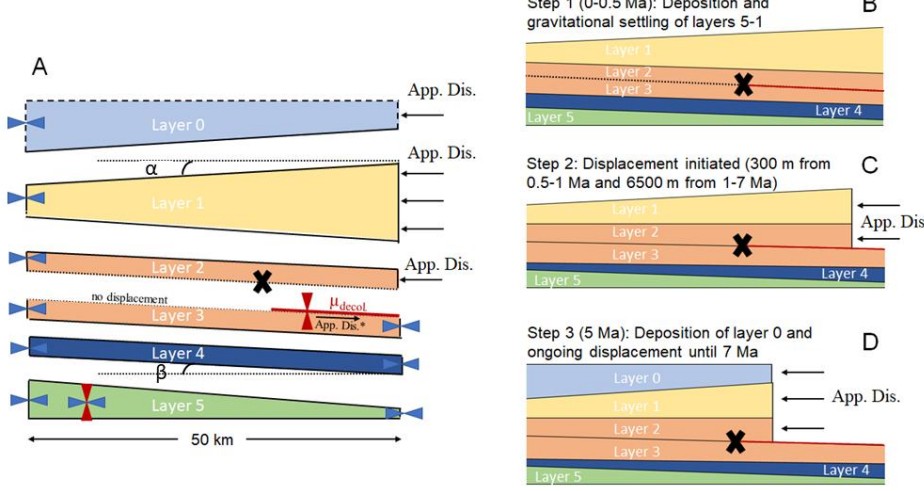


**Figure 1: Initial geometry, boundary conditions, displacement and modelling steps which are used during simulation. A: Initial model geometry and boundary conditions. μ$_{decol.}$ and App. Dis. stands for friction coefficient of the décollement and applied displacement respectively, while β and α shows the dip angle of the décollement and initial taper surface, respectively. App. Dis.\* indicates the additionally tested displacement condition where sliding across the**
**décollement is fixed at the same rate as the lateral displacement. B-D: Key modelling steps with initial gravitational settling (B), displacement at two different rates (C) and deposition of layer 0 (D). The black cross symbol shows the initial location of the discontinuity point over the décollement.**

**2.3 Material constitutive behavior and properties**

We use the SR3 constitutive model (Crook et al., 2006) which is based on critical state soil mechanics (Wood, 1990). Volumetric changes in critical state soil mechanics consider both mean effective stress p and deviatoric stress q in deformation related sediment compaction (Figure 2). Mean effective stress p is calculated as follows:

$$p' = \frac{\sigma'_1 + \sigma'_2 + \sigma'_3}{3} = \frac{\sigma'_h + \sigma'_H + \sigma'_v}{3} \qquad\qquad \text{Eq.2}$$


And deviatoric stress q results from:

$$q = \sqrt{\frac{(\sigma_1 - \sigma_2)^2 + (\sigma_2 - \sigma_3)^2 + (\sigma_3 - \sigma_1)^2}{2}} \qquad\qquad \text{Eq.3}$$

In equation 2 and equation 3 $\sigma_1, \sigma_2, \sigma_3$ and $\sigma'_1, \sigma'_2, \sigma'_3$ represent the maximum, medium and minimum principal
stresses and principal effective stresses, which can be either the minimum horizontal stress $\sigma_h$, the vertical stress $\sigma_v$ or maximum horizontal stress $\sigma_H$, respectively.

The SR3 yield surface, is divided by the critical state line into a compaction side representing diffuse strain and a dilation part corresponding to strain localization. The dip angle of the critical state line is calculated as (cf. Gao et al. 2018):


$$\eta_{cs} = \tan \xi \left[ (n_{sr3} + 1)^{-\frac{1}{n_{sr3}}} \right]$$  Eq. 4

Where $\xi$ is a friction parameter which is closely related to the internal friction angle and $n_{sr3}$ is a material constant.

The yield surface intersects the mean effective stress line at two points which define the tensile intersection (pt) and the pre-consolidation pressure (pc) of the material. On the yield surface, porosity is constant and can therefore change both as a function of mean effective stress and deviatoric stress.

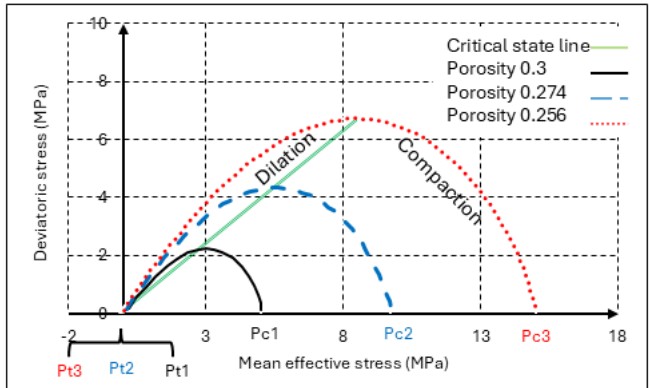

**Figure 2: Yield surface in SR3 model (for the case with pre-consolidation pressure of 5 MPa). The blue dashed line and red dotted lines indicate the change of the yield surface with mean effective stress increasing to 10 MPa and 15 MPa, respectively.**

The deformation inside the yield surface follows an elastic behaviour, which is dependent on Young's modulus and Poisson's ratio ($\nu$, constant in this study), and the following equation is utilized to characterize this

deformation:

$$E = E_{ref} \left[ \frac{\sigma_3 + A}{B} \right]^{n_e}$$  Eq. 5

Where $E$, $E_{ref}$, $\sigma_3$, $A$, $B$ and $n_e$ shows Young's modulus, reference Young's modulus, the minimum principal

stress, 1st material constant, 2nd material constant and shape parameter respectively.

In addition, we assign a constant grain density $\rho_s$ to each layer, which constrains vertical stress evolution. Table 1 summarizes all constant material properties and the values used in each layer.

**Table 1: SR3 material properties for the model layers.**

| Property | Layer 5 | Layer 4 | Remaining layers | Units |
|---|---|---|---|---|
| Reference Young's modulus ($E_{ref}$) | 45 | 25 | 5 | $GPa$ |

| | | | | |
|---|---|---|---|---|
| Poisson ratio ($\nu$) | 0.25 | 0.25 | 0.25 | - |
| Material constant ($A$); a constant to ensure smooth transition through near zero effective mean stress | -0.28 | -0.28 | -0.28 | - |
| Material constant ($B$); a constant to ensure smooth transition through near zero effective mean stress | -0.28 | -0.28 | -0.28 | - |
| Grain density ($\rho_s$) | 2700 | 2700 | 2400 | $kg/m^3$ |
| Initial pre-consolidation pressure ($pc$) | Table 2 | Table 2 | Table 2 | $MPa$ |
| Initial tensile intercept ($pt$) | -0.01×$pc$ | -0.01×$pc$ | -0.01×$pc$ | $MPa$ |
| Friction parameter ($\xi$ in Eq.2) | 56 | 56 | 56 | $^\circ$ |
| Material constant ($ne$); $E$ dependency on the minimum principal stress | 0.3 | 0.3 | 0.3 | - |
| Material constant ($n_{sr3}$) | 1.3 | 1.3 | 1.3 | - |

**2.4 Sensitivity analysis cases**

We investigate the impact of material strength, décollement dip angle and décollement friction on back-thrust development in early (after 2 Ma) and late (after 7 Ma) deformation stages. We vary the initial pre-consolidation pressure $pc$ between 2.5 and 50 MPa and the initial tensile intercept $pt$ as a function of $pc$ with $pt = -0.01 \times pt$ as a proxy for material strength (higher pc = higher material strength). The friction coefficient of the décollement ($\mu_{decol.}$ on red line to the right of X in Figure 1) is varied between 0 and 0.3. The taper angles ($\beta$ = dip of décollement; $\alpha$ = initial dip of surface layer 1; cf. Figure 1) are varied between 2° to 6° and 0° to -2°, respectively. In total, the variations of the described parameters result in 36 models, however, 6 models are also run with a controlled displacement rate across the décollement coupled to the lateral displacement rate. Table 2 summarizes all model scenarios.

We then categorize the simulation results after 2 Ma and 7 Ma into the formation of no/negligible (N, rank =1), weak (Y-W, rank =2) or strong back-thrusting (Y-S, rank =3). To do so, maximum effective plastic strain (the plastic component of the rate of deformation tensor) in the back-thrust is used as the criteria for these categories. Maximum effective plastic of 0.1 and 0.5 is used for boundary between N, Y-W and Y-S respectively at early times (2 Ma), while 0.5 and 1 is used at late times (7 Ma). A Spearman rank correlation is then performed to assess the impact of the varied input parameters: décollement dip, décollement friction coefficient and pre-consolidation pressure (material strength). Note that we do not assess the impact of the initial surface angle, because we coupled it to the décollement dip (cf. Table 2). To perform the rank correlation, we also rank the input parameters décollement dip, décollement friction coefficient and pre-consolidation pressure (material strength) in ascending order (e.g. pre-consolidation pressure of 2.5 MPa becomes rank 1, while pre-consolidation pressure of 50 MPa becomes rank 4; cf. Table 2) and then calculate Pearson's coefficient of correlation between the back-thrusting ranks and the ranks of the varied input parameters.

**3 Results**

The results of all simulation cases in Table 2, along with the cases involving constant bottom boundary displacement, are analyzed at three timesteps: 2 Ma, 4 Ma, and 7 Ma. Figures of these time steps showing the distribution of plastic strain, horizontal-to-vertical stress ratio $\sigma_{xx}/\sigma_{zz}$, maximum shear stress-to-mean effective

stress ratio $\tau_{max}/p$ and porosity $n$ for all modelled cases are provided in the supplementary section B. End-member cases are specifically selected for detailed examination.


**Table 2: Input parameters and results of the sensitivity analysis. In cases depicted with a star, additional simulations are conducted where the displacement along the décollement is coupled to the lateral displacement rate, which we apply at the right-hand side of the models. The notations Y-S, Y-W, and N represent development of strong, weak, and no or negligible back-thrusts, respectively.**

| Case # | Model definition | | | | Results | |
|---|---|---|---|---|---|---|
| | Décollement dip β (°) | Initial surface (layer 1) dip of wedge α (°) | Friction coefficient of décollement μ_decol. | Material strength / pre-consolidation pressure pc (MPa) | Early (2 Ma) back-thrusting category | Late (7 Ma) back-thrusting category |
| 1 (Figure 3) | 6 | 0 | 0 | 2.5 | Y-W | Y-W |
| 2* | 6 | 0 | 0 | 5 | Y-W | Y-W |
| 3* | 6 | 0 | 0 | 10 | Y-W | Y-W |
| 4 (Figure 4) | 6 | 0 | 0 | 50 | Y-S | Y-S |
| 5 (Figure 5) | 6 | 0 | 0.15 | 2.5 | N | Y-W |
| 6* | 6 | 0 | 0.15 | 5 | N | Y-W |
| 7* (Figure 8) | 6 | 0 | 0.15 | 10 | Y-W | Y-W |
| 8 | 6 | 0 | 0.15 | 50 | Y-S | Y-S |
| 9 | 6 | 0 | 0.3 | 2.5 | N | N |
| 10 | 6 | 0 | 0.3 | 5 | N | N |
| 11 (Figure 6) | 6 | 0 | 0.3 | 10 | Y-W | N |
| 12 | 6 | 0 | 0.3 | 50 | Y-S | Y-S |
| 13 | 4 | -1 | 0 | 2.5 | Y-W | Y-W |
| 14 | 4 | -1 | 0 | 5 | Y-W | Y-W |
| 15 | 4 | -1 | 0 | 10 | Y-S | Y-W |
| 16 (Figure 7) | 4 | -1 | 0 | 50 | Y-S | Y-S |
| 17 | 4 | -1 | 0.15 | 2.5 | N | Y-W |
| 18 | 4 | -1 | 0.15 | 5 | Y-W | Y-W |
| 19 | 4 | -1 | 0.15 | 10 | Y-S | Y-W |
| 20 | 4 | -1 | 0.15 | 50 | Y-S | Y-S |
| 21 | 4 | -1 | 0.3 | 2.5 | N | N |
| 22 | 4 | -1 | 0.3 | 5 | Y-W | N |
| 23 | 4 | -1 | 0.3 | 10 | Y-W | N |
| 24 | 4 | -1 | 0.3 | 50 | Y-S | Y-S |
| 25 | 2 | -2 | 0 | 2.5 | Y-W | Y-W |
| 26 | 2 | -2 | 0 | 5 | Y-W | Y-W |
| 27 | 2 | -2 | 0 | 10 | Y-S | Y-W |
| 28 | 2 | -2 | 0 | 50 | Y-S | Y-S |
| 29 | 2 | -2 | 0.15 | 2.5 | N | Y-W |
| 30 | 2 | -2 | 0.15 | 5 | N | Y-W |
| 31 | 2 | -2 | 0.15 | 10 | Y-S | Y-W |
| 32 | 2 | -2 | 0.15 | 50 | Y-S | Y-S |
| 33 | 2 | -2 | 0.3 | 2.5 | N | N |
| 34* | 2 | -2 | 0.3 | 5 | N | N |
| 35* | 2 | -2 | 0.3 | 10 | Y-W | N |

| 36 | 2 | -2 | 0.3 | 50 | Y-S | Y-S |
|----|---|----|-----|----|-----|-----|


### 3.1 Material strength (pre-consolidation pressure) *pc*

Figure 3 illustrates the progressive development of plastic strain and the final values of the $\sigma_{xx}/\sigma_{zz}$ and $\tau_{max}/p$ for Case #1 ($\beta = 6°, \alpha = 0°, \mu_{decol.} = 0,$ and pc = 2.5 MPa), which represents high décollement dip angle, low décollement friction coefficient and low material strength (Table 2).

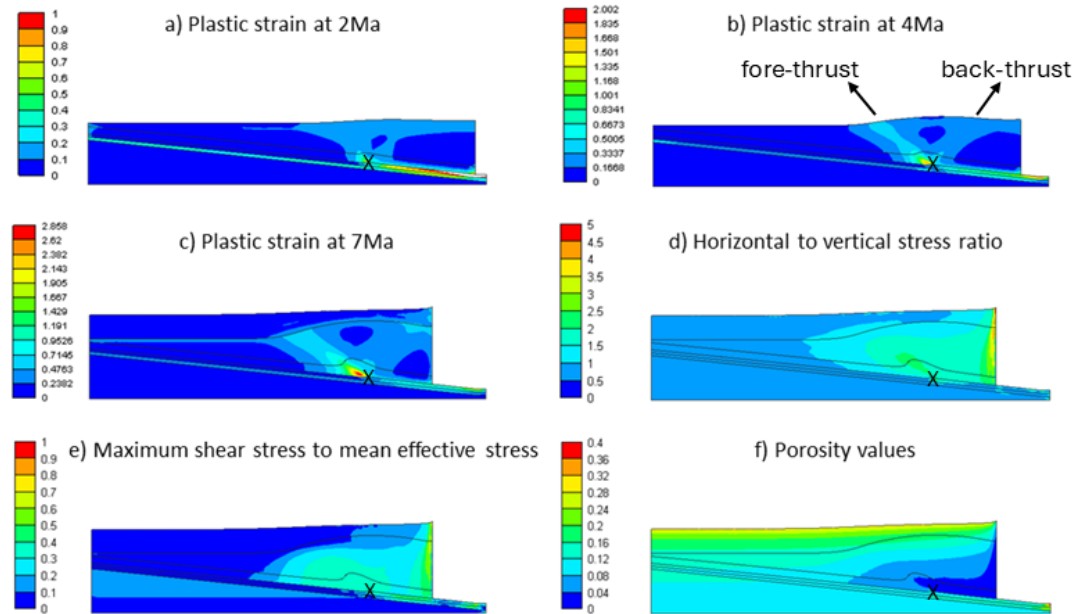


**Figure 3: Results of Case #1 ($\beta = 6°, \alpha = 0°, \mu_{decol.} = 0,$ and pc = 2.5 MPa). A-C: Progressive development of plastic strain at 2 Ma, 4 Ma and 7 Ma, respectively. D-F: Distribution of horizontal-to-vertical stress ratio $\sigma_{xx}/\sigma_{zz}$, maximum shear stress-to-mean effective stress ratio $\tau_{max}/p$, and porosity $n$ at 7 Ma. The black cross symbol shows the initial location of the discontinuity point over the décollement.**


In Case #1, due to the assumed zero-friction coefficient along the décollement, fault initiation occurs from the discontinuity point, and up to a certain point, overburden layers slide easily over the décollement. At 2 Ma (Figure 3A), deformation is predominantly diffusive, resulting in a less efficient fore-thrust (it does not reach to the surface). Additional displacement up to 4 Ma (Figure 3B) results in the development of a strong fore-thrust that reaches the surface, accompanied by a weak conjugated back-thrust. From 5 Ma to 7 Ma, simultaneous sedimentation is implemented with displacement. The developed fore-thrust and back-thrust do not extend through the later sedimented materials (Figure 3C). Elevated $\tau_{max}/p$ ratio is observed near to the discontinuity point and areas close to the displacing boundary (Figure 3E), while the $\sigma_{xx}/\sigma_{zz}$ (Figure 3D) reveals the dominance of horizontal stress in the wedge area. Since porosity is controlled by mean stress and deviatoric stress in critical state soil mechanics, and materials in Case #1 have a low pre-consolidation pressure (weak material), these areas show a higher porosity loss compared to their counterparts at the same depth in the foreland (Figure 3F). Additionally, it should be noted that weak material and a low basal friction coefficient in this scenario result in a low final dip of the wedge surface.

From Case #1 to Case #4, the initial geometry and friction coefficients on the décollement remain constant, while the pre-consolidation pressure increases from 2.5 MPa in Case #1 to 5, 10, and 50 MPa in cases #2, #3, and #4, respectively (materials become stronger from mechanical point of view). In all these cases, due to the zero-friction coefficient along the décollement, overburden layers slide over the décollement, and thrusting initiates from the discontinuity point. With an increase in material strength, thrusts become more defined. Additionally, the

increasing material strength leads to a higher final dip angle of the wedge surface. Moreover, the area with higher $\sigma_{xx}/\sigma_{zz}$ increases in size. Comparing Case #1 ($\beta = 6°, \alpha = 0°, \mu_{decol.} = 0$, and pc $= 2.5$ MPa) to Case #2 ($\beta = 6°, \alpha = 0°, \mu_{decol.} = 0$, and pc $= 5$ MPa) reveals a more developed back-thrust in the latter. This trend continues, and Case #3 ($\beta = 6°, \alpha = 0°, \mu_{decol.} = 0$, and pc $= 10$ MPa) shows a fully developed back-thrust and a weak second back-thrust at a later time. Figure 4 illustrates the results of Case #4 ($\beta = 6°, \alpha = 0°, \mu_{decol.} = 0$, and pc $=$

50 MPa), where strong back-thrusts developed.

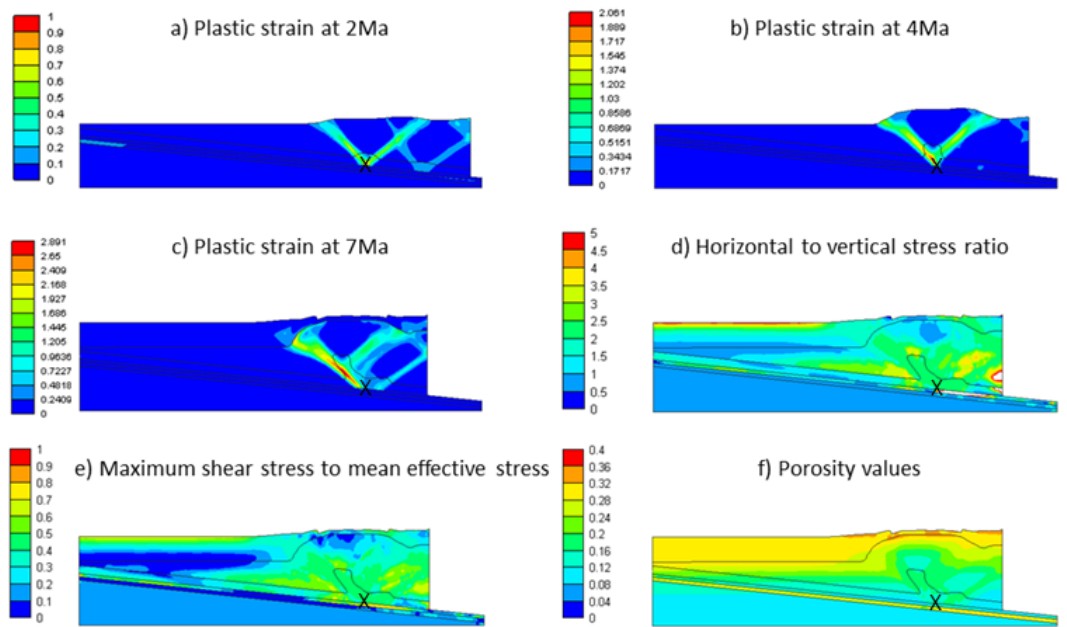

**Figure 4: Progressive development of plastic strain and the final values of the $\sigma_{xx}/\sigma_{zz}$ , $\tau_{max}/p$ and *n* for the case #4 ($\beta =$**
**$6°, \alpha = 0°, \mu_{decol.} = 0$, and pc $= 50$ MPa). The black cross symbol shows the initial location of the discontinuity point over the décollement.**

Case #4 ($\beta = 6°, \alpha = 0°, \mu_{decol.} = 0$, and pc $= 50$ MPa) exhibits two sets of conjugated fore-thrust and back-thrust development in early times (Figure 4A-C). Similar behavior is observed in corresponding cases with

different dip angles (Case #16: $\beta = 4°, \alpha = -1°, \mu_{decol.} = 0$, and pc $= 50$ MPa & Case #18: $\beta = 4°, \alpha = -1°, \mu_{decol.} = 0.15$, and pc $= 5$ MPa). Also, Case #11 ($\beta = 6°, \alpha = 0°, \mu_{decol.} = 0.3$, and pc $= 10$ MPa), #23 ($\beta = 4°, \alpha = -1°, \mu_{decol.} = 0.3$, and pc $= 10$ MPa), and #35 ($\beta = 2°, \alpha = -2°, \mu_{decol.} = 0.3$, and pc $= 10$ MPa) show a similar deformation with weak thrusts, and these are corresponding cases that differ in dip angles as well. As the deformation progresses, one set of these conjugated thrusts develops further, and the other

disappears. In late times, two back-thrusts and one fore-thrust are fully developed, and there is a weak back-thrust. High values of $\tau_{max}/p$ expand in larger areas of the model (Figure 4E). The $\sigma_{xx}/\sigma_{zz}$ is higher than in Case #1 ($\beta = 6°, \alpha = 0°, \mu_{decol.} = 0$, and pc $= 2.5$ MPa) with lower pre-consolidation pressure (Figure 4D). In this case, layers

resist porosity loss, and porosity remains high in the thrusted area. Areas highly affected by horizontal stress are evident in Figure 4F.

**3.2 Friction coefficient of the décollement $\mu_{decol.}$**

Cases #5-8 ($\beta = 6°, \alpha = 0°, \mu_{decol.} = 0.15$) are similar to cases #1-4 ($\beta = 6°, \alpha = 0°, \mu_{decol.} = 0$), except for an increased friction coefficient of the décollement $\mu_{decol.} = 0.15$. The simulation results for Case #5 ($\beta = 6°, \alpha = 0°, \mu_{decol.} = 0.15, \text{and pc} = 2.5$ MPa) from this series are depicted in Figure 5A-F. In the early stages, a weak-thrust development near the displacing boundary condition is evident, accompanied by the initiation of a fore-thrust at the discontinuity (Figure 5A-C). It should be noted that in this case, friction along the décollement imposes restrictions on free sliding, resulting in deformations near the displacing boundary condition. As time progresses, the initiation of a weak back-thrust becomes visible near the discontinuity, although it disappears in the late stages. During this period, deformation is primarily controlled by diffusive plastic strain (Figure 5A-C). Notably, the forces required to achieve the same displacement as in Case #1 ($\beta = 6°, \alpha = 0°, \mu_{decol.} = 0, \text{and pc} = 2.5$ MPa) are higher, leading to increased $\sigma_{xx}/\sigma_{zz}$ (Figure 5E). Porosity loss in areas unaffected by thrusting is more pronounced compared to Case #1 (Figure 5F). This pattern is consistently observed across corresponding cases #5-8 and #1-4, with thrusts developed in Cases #5-8 generally exhibiting weaker characteristics. End member of this series (Case #8: $\beta = 6°, \alpha = 0°, \mu_{decol.} = 0.15, \text{and pc} = 50$ MPa) exhibits the presence of two fully-developed back-thrusts. In contrast, Case #4 ($\beta = 6°, \alpha = 0°, \mu_{decol.} = 0, \text{and pc} = 50$ MPa) displays two fully-developed back-thrusts alongside a weaker back-thrust.

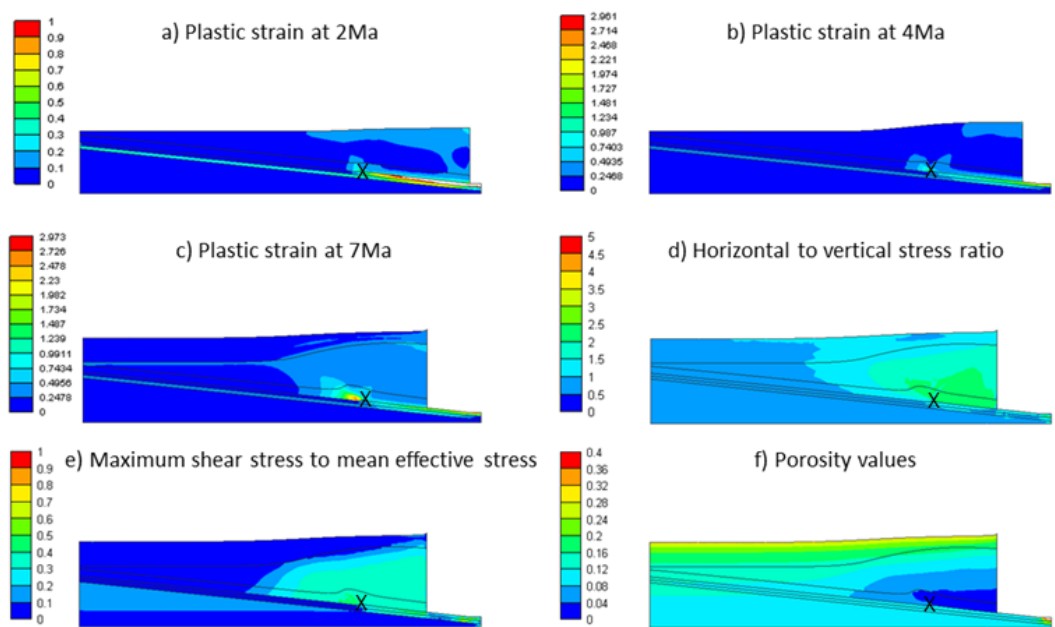

**Figure 5: Progressive development of plastic strain and the final values of the $\sigma_{xx}/\sigma_{zz}$, $\tau_{max}/p$ and n for the case #5 ($\beta = 6°, \alpha = 0°, \mu_{decol.} = 0.15, \text{and pc} = 2.5$ MPa). The black cross symbol shows the initial location of the discontinuity point over the décollement.**

In Cases #9-12 ($\beta = 6°, \alpha = 0°, \mu_{\text{decol.}} = 0.3$), the friction coefficient of the décollement is further increased to $\mu_{\text{decol.}} = 0.3$. Within this series, deformation is not restricted to the vicinity of the discontinuity point; also, a significant portion of it occurs near the displacing boundary (Figure 6A-C as an example for Case #11: $\beta = 6°, \alpha = 0°, \mu_{\text{decol.}} = 0.3, \text{and pc} = 10 \text{ MPa}$). This results in a steeper slope of deformation on the surface of the top layer. In contrast to the earlier series (cases #1-4: $\beta = 6°, \alpha = 0°, \mu_{\text{decol.}} = 0$ and #5-8: $\beta = 6°, \alpha = 0°, \mu_{\text{decol.}} = 0.15$), where fore-thrusting near the discontinuity point led to elevated $\sigma_{xx}/\sigma_{zz}$ features dipping towards the top in the direction of fore-thrusting (see Figure 3d-5d), Cases #9-12 ($\beta = 6°, \alpha = 0°, \mu_{\text{decol.}} = 0.3$) exhibit horizontally aligned stress plumes. These cases demonstrate higher porosity loss in weak materials near the displacing boundary. Thrusting in this series is observed to be weaker compared to the previous two sets, and the likelihood of back-thrusting decreases. For instance, Case #10 ($\beta = 6°, \alpha = 0°, \mu_{\text{decol.}} = 0.3, \text{and pc} = 5 \text{ MPa}$) displays two early-stage fore-thrusts, while Case #2 ($\beta = 6°, \alpha = 0°, \mu_{\text{decol.}} = 0, \text{and pc} = 5 \text{ MPa}$) presents a set of conjugated fore-thrusts and back-thrusts. Figure 6 illustrates Case #11 ($\beta = 6°, \alpha = 0°, \mu_{\text{decol.}} = 0.3, \text{and pc} = 10 \text{ MPa}$) from this series, clearly demonstrating the influence of the friction coefficient on deformation development near the displacing boundary. It exhibits one fully developed fore-thrust and another in the early stage of development. In contrast, Case #3 ($\beta = 6°, \alpha = 0°, \mu_{\text{decol.}} = 0, \text{and pc} = 10 \text{ MPa}$) shows a fully developed back-thrust and a weak thrust at a later time.

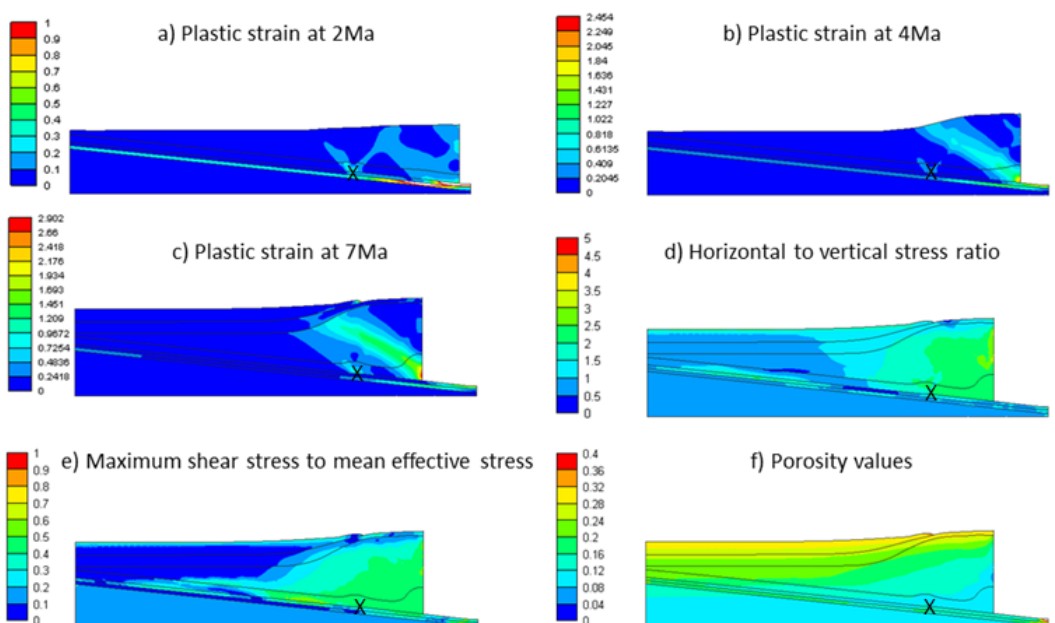

**Figure 6: Progressive development of plastic strain and the final values of the $\sigma_{xx}/\sigma_{zz}$, $\tau_{max}/p$ and n for the case #11 ($\beta = 6°, \alpha = 0°, \mu_{\text{decol.}} = 0.3, \text{and pc} = 10 \text{ MPa}$). The black cross symbol shows the initial location of the discontinuity point over the décollement.**

### 3.3 Décollement dip angle β

Cases #13-24 ($\beta = 4°, \alpha = -1°$) were designed to examine the influence of décollement dip angles (β) on deformation. When comparing the deformation results of cases #13-16 ($\beta = 4°, \alpha = -1°, \mu_{\text{decol.}} = 0$) with their corresponding counterparts in #1-4 ($\beta = 6°, \alpha = 0°, \mu_{\text{decol.}} = 0$), minimal differences are observed in cases with weak material (#1-3 and #13-15). However, significant differences emerge between cases #4 ($\beta = 6°, \alpha = 0°, \mu_{\text{decol.}} = 0, \text{and pc} = 50 \text{ MPa}$) and #16 ($\beta = 4°, \alpha = -1°, \mu_{\text{decol.}} = 0, \text{and pc} = 50 \text{ MPa}$), with strong

material. Figure 7 illustrates the deformation progression in Case #16 ($\beta = 4°, \alpha = -1°, \mu_{decol.} = 0,$ and pc $=$ 50 MPa). In comparison to Case #4 ($\beta = 6°, \alpha = 0°, \mu_{decol.} = 0,$ and pc $=$ 50 MPa), it shows three fully developed back-thrusts and a set of conjugated fore-thrusts and back-thrusts at a later time (Figure 7c). Additionally, the $\sigma_{xx}/\sigma_{zz}$ is higher near the discontinuity point. Similar behavior is noted for cases in the range of #17-24 ($\beta = 4°, \alpha = -1°$) when compared to their corresponding cases in #5-12 ($\beta = 6°, \alpha = 0°$). This implies

that cases with weak material exhibit negligible differences, while end members with strong material (cases #20: $\beta = 4°, \alpha = -1°, \mu_{decol.} = 0.15,$ and pc $=$ 50 MPa and #24: $\beta = 4°, \alpha = -1°, \mu_{decol.} = 0.3,$ and pc $=$ 50 MPa) display considerable differences in comparison to cases #8 ($\beta = 6°, \alpha = 0°, \mu_{decol.} = 0.15,$ and pc $=$ 50 MPa) and #12 ($\beta = 6°, \alpha = 0°, \mu_{decol.} = 0.3,$ and pc $=$ 50 MPa). Notably, back-thrusting in cases #20 ($\beta = 4°, \alpha = -1°, \mu_{decol.} = 0.15,$ and pc $=$ 50 MPa) and #24 ($4 = 6°, \alpha = -1°, \mu_{decol.} = 0.3,$ and pc $=$ 50 MPa) is stronger.


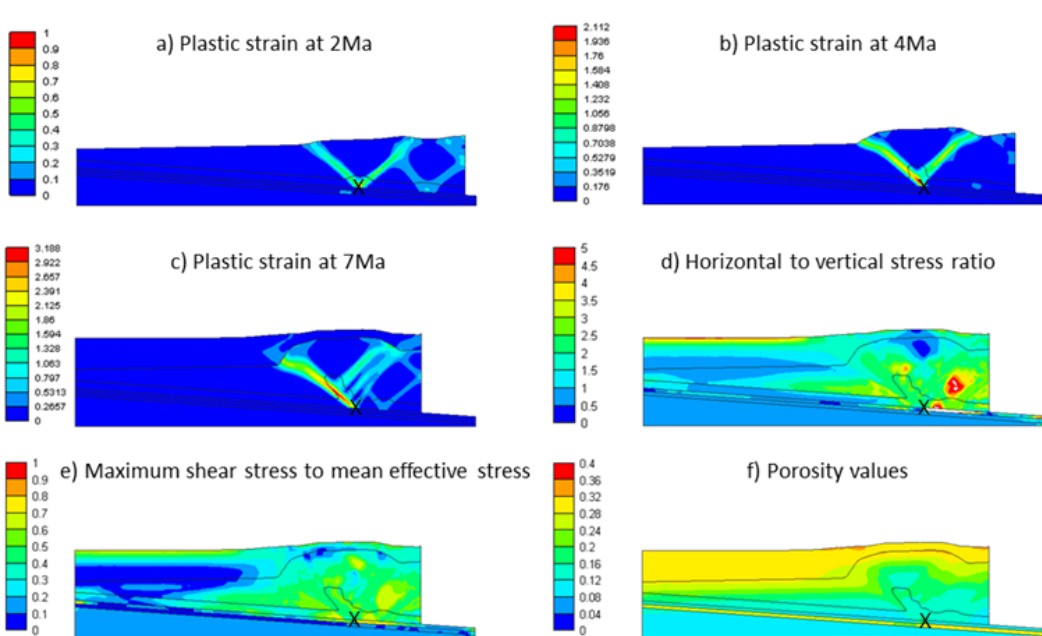

**Figure 7: Progressive development of plastic strain and the final values of the σ$_{xx}$/σ$_{zz}$, τ$_{max}$/p and n for the case #16 ($\beta =$ 4°, $\alpha = -1°, \mu_{decol.} = 0,$ and pc $=$ 50 MPa). The black cross symbol shows the initial location of the discontinuity point**
**over the décollement.**

Cases #25-36 ($\beta = 2°, \alpha = -2°$) were designed to explore additional dip angle scenarios, wherein the décollement dip angle becomes even smoother, and the dip angle of the top surface becomes even deeper. Similar to Cases #13-24 ($\beta = 4°, \alpha = -1°$), differences between this series and Cases #1-12 ($\beta = 6°, \alpha = 0°$) are

negligible for weak materials, while for cases with strong material, differences are considerable. Case #28 ($\beta = 2°, \alpha = -2°, \mu_{decol.} = 0,$ and pc $=$ 50 MPa), for instance, reveals one fully developed fore-thrust with two back-thrusts intersecting it. Additionally, there is a set of conjugated fore-thrusts and back-thrusts within the distance between the displacing boundary and the discontinuity point. Cases #8 ($\beta = 6°, \alpha = 0°, \mu_{decol.} = 0.15,$ and pc $=$ 50 MPa) and #32 ($\beta = 2°, \alpha = -2°, \mu_{decol.} = 0.15,$ and pc $=$ 50 MPa) exhibit smaller differences. Notably,

Case #36 ($\beta = 2°, \alpha = -2°, \mu_{decol.} = 0.3,$ and pc $=$ 50 MPa) illustrates the thrusts observed in Case #12 ($\beta =$

$6°, \alpha = 0°, \mu_{decol.} = 0.3,$ and pc $= 50$ MPa) at a later time, along with a fore-thrust rooted from the displacing boundary.

### 3.4 Displacing boundary condition

Up to this point, the displacing boundary condition has been exclusively assigned to the right boundary of the overburden layers. To examine the impact of these boundary conditions on deformation development, certain cases (indicated with * in the Table 2) assume that the bottom of the overburden layers on the right section of the discontinuity point moves at the same displacement rate as the side wall. The results for these cases are presented in the Supplementary B, and Case #7* ($\beta = 6°, \alpha = 0°, \mu_{decol.} = 0.15,$ and pc $= 10$ MPa) is chosen for discussion as a representative example (see Figure 8). In these cases, a more pronounced back-thrust is observed compared to the developed fore-thrusts. Additionally, the dip of the highly deformed section in layer 2 is oriented towards the top and right, in contrast to the previous boundary condition where it dipped towards the left and top. Furthermore, areas characterized by high $\sigma_{xx}/\sigma_{zz}$ and $\tau_{max}/p$ propagate farther into the foreland, occupying larger areas. Despite the same material properties, porosity losses are comparable in both situations.

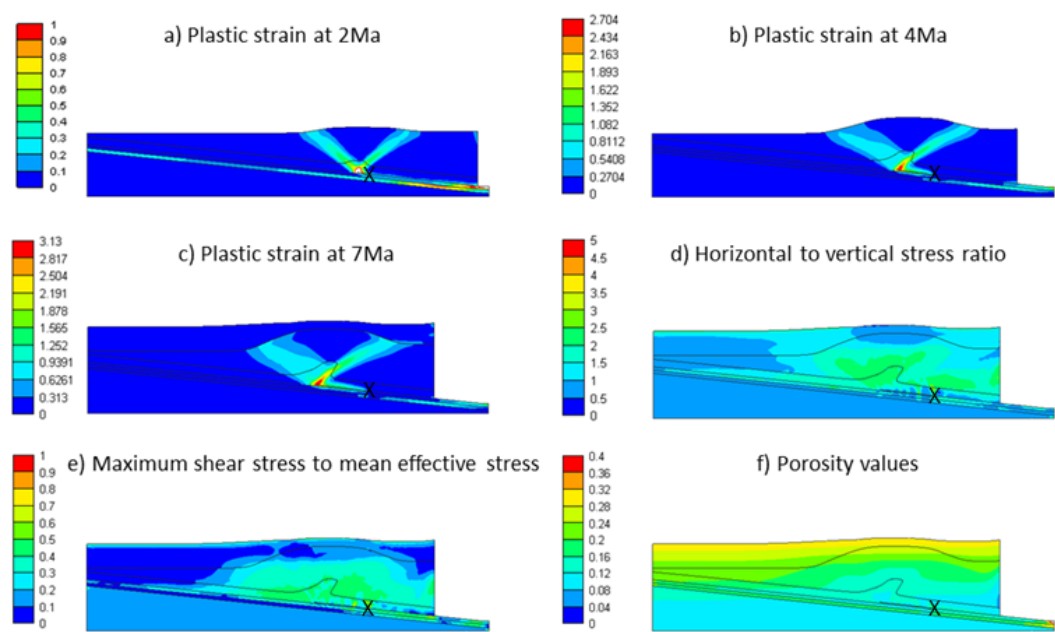

**Figure 8: Progressive development of plastic strain and the final values of the $\sigma_{xx}/\sigma_{zz}$, $\tau_{max}/p$ and n for the case #7* ($\beta = 6°, \alpha = 0°, \mu_{decol.} = 0.15,$ and pc $= 10$ MPa). The black cross symbol shows the initial location of the discontinuity point over the décollement.**

## 4 Discussion

### 4.1 Impact of material strength

Starting with Case #1 ($\beta = 6°, \alpha = 0°, \mu_{decol.} = 0,$ and pc $= 2.5$ MPa), which involves weak material and a low basal friction coefficient, the resulting low dip of the wedge at the surface aligns with critical taper theory. The progressive development of deformation in Case #1 (Figure 3) reveals that thrusting cannot penetrate the late sedimented material. This limitation can be explained by the fact that all late-sedimented materials can only

undergo under-consolidation processes (or follow normal consolidation), restricting their deformation to a ductile manner. In this context and based on the SR3 model, thrusting in these sediments is plausible if other compaction mechanisms than the mechanical one contribute. These compaction mechanisms enhance the strength of the newly deposited material and bring them to the over-consolidated condition (Obradors-Prats et al., 2017).

Advancing from Case #1 ($\beta = 6°, \alpha = 0°, \mu_{decol.} = 0$, and pc = 2.5 MPa) to Case #4 ($\beta = 6°, \alpha = 0°, \mu_{decol.} = 0$, and pc = 50 MPa), the pre-consolidation pressure increases. As reported by Obradors-Prats et al. (2017), in a highly consolidated state, the likelihood of the stress path intersecting the yield curve on the shear section is higher. This intersection is crucial for modelling brittle deformation, a form of plastic deformation predictable with Cam clay-type models. Such models result in fast softening materials, eventuate to localization behavior

characterized by steep displacement gradients in thin zones of strong plastic strain. Furthermore, in accordance with critical taper theory, the increasing material strength results in a higher dip angle of the surface layer. The increase in the $\tau_{max}/p$ brings the system closer to the critical state. Consequently, with the rise in material strength, the areas of material in a critical state increase in size (refer to the figures in the supplementary material). For cases #1-4, where the friction coefficient over the décollement is zero, and in cases with lower displacement

resistance at the basal layer/overburden layer interface, a minor rise in shear stress causes deformation at areas close to the discontinuity. Therefore, deformation can happen in distances far away from the displacing boundary. Conversely, in cases with higher resistance to displacement, brittle deformation tends to develop near the moving side walls.

**4.2 Impact of the décollement friction coefficient**

     As the friction coefficient increases in cases #5-8 ($\beta = 6°, \alpha = 0°, \mu_{decol.} = 0.15$) and #9-12 ($\beta = 6°, \alpha = 0°, \mu_{decol.} = 0.3$) in comparison to cases #1-4 ($\beta = 6°, \alpha = 0°, \mu_{decol.} = 0$), the dip angle of the surface of the top layer increases, a phenomenon previously observed in a study by Gao et al. (2018). The results of cases #1-12 ($\beta = 6°, \alpha = 0°$) align with previous studies indicating that low basal friction coefficient facilitates back-

thrusting, while high basal friction coefficient favor fore-thrusts (Seely, 1977; Davis and Engelder, 1985). The models proposed by van Hagke et al. (2023) also support this notion. van Hagke et al. (2023) noted that low basal friction coefficient alone is insufficient to systematically describe back-thrusting. Additionally, they highlighted that a very low basal dip (less than 0.5 degrees) is necessary for back-thrust dominance. Strong décollements (high basal friction coefficients), on the other hand, develop increased taper angles, changing the stress field in a way

that fore-thrusting becomes more favourable. This observation aligns with sand-box experiments in which highly back-thrusted systems developed in low angle basal with zero or low basal friction coefficients (Gutscher et al., 2001). However, other researchers argue that for low angle basal décollement, fore-thrusts may dominate (MacKay, 1995; Cotton and Koyi, 2000). The work of fault growth is examined through the experimental and simulation tools to shed light on this issue (Herbert et al., 2015; McBeck et al., 2017). Results demonstrated the

importance of the height of the sedimented layers and basal features, with fore-thrust slip influencing back-thrust development. The parametric study by van Hagke et al. (2023) showed that in flat, low-angle basal settings, both back-thrusting and fore-thrusting are possible, and the basal friction coefficient controls the dominance of one over the other. They suggest that for the low friction basal, back-thrusting requires lower work than fore-thrusting. Therefore, the chance of back-thrust dominancy is higher in such systems.

The discovery of back-thrust dominance in systems with a weak décollement and a low dip basal is intriguing. Previously, it was suggested that the likelihood of back-thrusting is higher in high-dip basal settings. Researchers claimed that in such systems, the direction of the maximum stress component and the basal décollement are nearly parallel. The back-thrust dip is gentle and, therefore, develops more easily (MacKay, 1995; Cubas et al., 2016). It should be noted that while these studies examine back-thrust dominance, the current study aims to identify the

conditions that facilitate back-thrust development. Additionally, there is a difference in the displacing boundary conditions between the current study and that of van Hagke et al. (2023). van Hagke et al. (2023) and other researchers (such as Obradors-Prats et al., 2017) assumed that the bottom of the top layers moves at the same displacement rate as the sidewall, whereas the current study primarily focuses on sidewall displacement. To investigate the impact of displacing boundary conditions, six additional simulations were conducted with a

moving bottom boundary of the overburden layers. With these considerations, back-thrusting was observed at a dip angle of 6°; however, a sensitivity analysis was not conducted to determine the end-member dip angle

**4.3 Impact of the décollement dip angle**

In analyzing the impact of the taper dip angle, while Case #1-3 ($\beta = 6°, \alpha = 0°, \mu_{\text{decol.}} = 0$) and #13-15 ($\beta =$

$4°, \alpha = -1°, \mu_{\text{decol.}} = 0.$) exhibit similar deformation patterns, Case #4 ($\beta = 6°, \alpha = 0°, \mu_{\text{decol.}} = 0,$ and pc $=$ 50 MPa) and #16 ($\beta = 4°, \alpha = -1°, \mu_{\text{decol.}} = 0,$ and pc $= 50$ MPa) display a distinct behavior. In these two cases, the material strength is high (50 MPa), and it appears that the top layers slide as a single solid part over the décollement. Consequently, the decrease in the dip angle of the décollement in Case #16 ($\beta = 4°, \alpha =$ $-1°, \mu_{\text{decol.}} = 0,$ and pc $= 50$ MPa) facilitates easier sliding over the décollement, with all restrictions

concentrating on the discontinuity point, resulting in more pronounced deformation for this case. It's noteworthy that the decrease in the dip angle of the décollement favors back-thrusting (as suggested by van Hagke et al. 2023), while the dip angle of the top surface restricts back-thrust development (as previously demonstrated that top surface erosion and flattening facilitate back-thrusting conditions, McClay, 2011). Similar to the comparison of Case #4 ($\beta = 6°, \alpha = 0°, \mu_{\text{decol.}} = 0,$ and pc $= 50$ MPa) & #12 ($\beta = 6°, \alpha = 0°, \mu_{\text{decol.}} = 0.3,$ and pc $=$

50 MPa), Case #20 ($\beta = 4°, \alpha = -1°, \mu_{\text{decol.}} = 0.15,$ and pc $= 50$ MPa) and #24 ($\beta = 4°, \alpha = -1°, \mu_{\text{decol.}} =$ $0.3,$ and pc $= 50$ MPa) exhibit stronger back-thrusting in comparison to Cases #8 ($\beta = 6°, \alpha = 0°, \mu_{\text{decol.}} =$ $0.15,$ and pc $= 50$ MPa) and #12 ($\beta = 6°, \alpha = 0°, \mu_{\text{decol.}} = 0.3,$ and pc $= 50$ MPa), respectively. While weak materials are not highly sensitive to the dip angle, the observations thus far lead to the conclusion that decreasing the dip angle of the décollement contributes to facilitating back-thrusting.

In the next step, the dip angle of the décollement is decreased ($\beta$ from 4° to 2°), while simultaneously the dip of the surface is increased ($\alpha$ from -1° to -2°), and the simulations are run as in Cases #25-36 ($\beta = 2°, \alpha = -2°$). Examination of the results indicates that while the decreasing décollement dip angle helps to enhance back-thrusting, the surface slope (top surface angle of layer 1) neutralizes some effects, resulting in weaker back-thrusting compared to the corresponding cases in Cases #13-24 ($\beta = 4°, \alpha = -1°$). With the consideration of the

higher surface dip, the system's activity in the hinterland, aimed at bringing the system to the critical taper state, decreases, leading to a decrease in the chance of back-thrusting.

In this research, the sedimentation and erosion processes are not examined in detail. However, McClay (2011) highlighted that a model excluding syn-contractional sedimentation or erosion demonstrated a general forward breaking sequence with simultaneous thrust activity. The inclusion of syn-contractional sedimentation resulted in

more extensive wedges characterized by reduced main forward converging thrusts and reduced thrust activities in the foreland. Conversely, syn-contractional erosion impeded the forward propagation of the deformation front, decreased quantity of main thrusts, and heightened thrust activities in the hinterland. When combined, the effects of syn-contractional sedimentation and erosion were found to be complementary. In the case of an eroded wedge approaching a subcritical taper state, shortening induces internal deformation, which results in a higher surface

slope. This adjustment enables the wedge to regain the critical state. The internal structural deformation within the thrust wedge contributes to an increased wedge thickness in the hinterland and may furthermore favor processes such as normal faulting in the inner wedge, back-thrust development, thrust reactivation, out-of-sequence thrust development, or basal accretion through duplexing at the base of the wedge.

### 4.4 Sensitivity analysis

**4.4.1 Impact of input décollement dip, décollement friction and material strength on back-thrusting**

The rank correlation analysis shows that pre-consolidation pressure has the largest impact on back-thrust development, both in early (after 2 Ma) and late (after 7 Ma) years (Figure 9). Thereby, the likelihood of back-thrusting increases with increasing pre-consolidation pressure, which we use as a proxy of material strength. Décollement friction coefficient is inversely related to back-thrust development. A weaker décollement increases

the likelihood of back-thrust development, and this effect seems to be slightly more important during the late stage of our simulations. Décollement dip is also inversely related to back-thrust development but only appears to have a minor effect, which, in our simulations, only has an influence during the early stages of deformation.

In summary, back-thrusting occurs if the contrast between the bulk strength and décollement strength is high (taper strength >> décollement strength). In other words, back-thrusting occurs, if sliding of the taper along the

décollement is easy, but the material of the taper is strong and pushes back. Accordingly, fore-thrusting is favored if sliding of the taper along décollement requires more work than shearing the taper (décollement detaches to new décollement resulting in a fore-thrust).

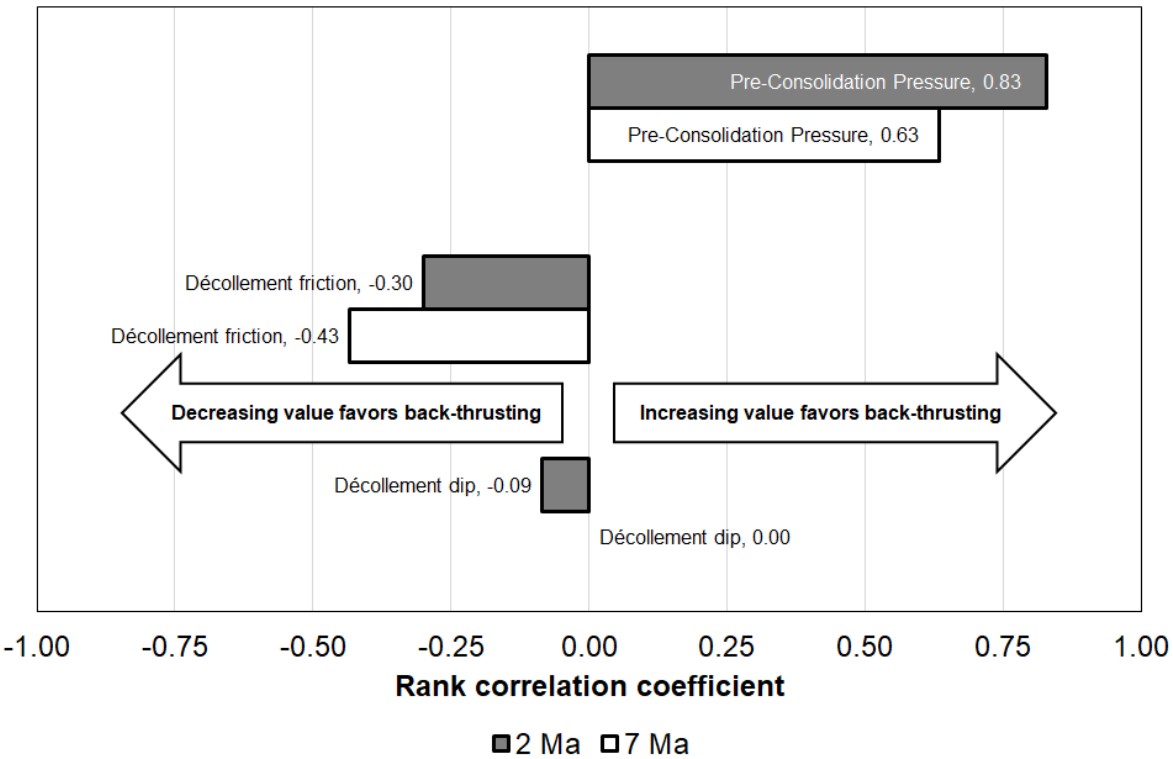

**Figure 9: Rank correlation analysis of the tested parameters impacting back-thrusting.**


**4.4.2 Geological processes influencing décollement dip, décollement friction and material strength**

Décollement dip angle probably is mostly controlled by pre-existing basement topography and the material strength of the footwall. In the course of ongoing deformation, the décollement dip probably loses its relevance to back-thrusting, because once sliding of the taper is initiated, the rate of sliding is likely only a function of the

frictional resistance along the décollement.

Décollement strength decreases if the resistance to slide along the décollement decreases, which is achieved if the décollement's friction coefficient and/or dip angle decrease. Factors controlling the décollement's friction coefficient are mainly the presence of low friction minerals such as clay minerals or evaporites (anhydrite, salt), pore pressure and diagenesis (e.g. Lebinson et al. 2020).

In our simulation, taper strength increases with its material strength (in our simulations = pre-consolidation pressure) similar to real-world analogs Geological processes which result in an increase of material strength or pre-consolidation pressure are burial and subsequent uplift or exhumation deposition of additional sediments during the deformation resulting in an increase of vertical (effective) stress and compaction and diagenesis, which can reduce porosity and again increase material strength.


**4.4 3 Limitations of the sensitivity analysis**

It should be noted that due to the computational cost of large strain geomechanical forward modelling the tested input parameter combinations are limited in this study. In addition, we categorized back-thrusting severity based

on the maximum effective plastic strain, but future studies could and probably should also include other measures such as displacement along fore- and back-thrusts and the number of back-thrusts developed.

Also, the performed rank correlation analysis only examines the individual influence of each individual input parameter on back-thrust formation. However, a multivariate analysis was deliberately avoided because the applied parameter combinations are probably not sufficient to allow for quantitative statements regarding the parameter dependencies on back-thrusting. The rank correlation results therefore should be understood as general trends regarding each input parameter's impact on back-thrusting and not as actual correlation coefficients.

The above-mentioned limitations point to a general lack of a standardized statistical approach to analyze modelling results of geodynamic modelling. Future studies should therefore aim at developing a standardized sensitivity analysis approach to allow for comparability between different studies and modelling methodologies.

**4.5 The role of boundary conditions in modelling back-thrust formation**

Fore- and back-thrusts are elementary structural components of fold-and-thrust belts and accretionary wedges, their formation has been intensively analyzed using numerical simulation (Buiter et al.; 2016), analogue modelling (Schreurs 2016) and structural field analysis (Lebinson et al. 2020). However, the main research focus has often been on fore-thrust formation as the main mechanism of hanging-wall accommodation. Although back-thrusts are very common in fold-and-thrust belts, they apparently account much less for total shortening compared to fore-thrusts. Buiter et al. (2016) show that in numerical simulation back-thrust formation is often associated with either a pop-up structure (one fore-thrust vs one back-thrust) or the stacking of fore-thrusts, where a back-thrust forms a steep antithetic ramp during stacking (forethrust >> backthrust); the authors also explain the role of basal friction on back-thrust formation where higher friction results in more fore-thrusts and vice versa (cf. Davis and Engelder 1985), which is in good agreement with our results. This observation is supported by Maillot & Leroy (2019) who explain how the dip of the ramp in combination with the friction coefficients of shear zones and back-thrusts control back-thrusting. van Hagke et al. (2023) reached similar conclusions in their literature review, highlighting that critical wedge experiments commonly exhibit fore-thrusts, with less frequent documentation of back-thrusting. When back-thrusts are present, they tend to form conjugates to fore-thrusts (Ellis et al., 2004). Field examinations on the fold-and-thrust-belt systems often show dominancy of fore-thrusts. Transitions from regions dominated by back-thrusts to those dominated by fore-thrusts can be observed in various geological settings, such as the European Alps (Ortner et al., 2023) or the Niger Delta (Higgins et al., 2009).

In real-world cases, the Cascadia fold-and-thrust belt (USA) could be an example of this category, where back-thrusts are the dominant form of deformation (Gutscher et al., 2001). To gain insights into the conditions leading to stronger back-thrusts than fore-thrusts, this study examines a different type of boundary condition, where the bottom boundary of the overburden layers moves at the same rate as the side boundary across the lower friction décollement. In this scenario, the strongest developed thrust is the back-thrust. In this condition, deformations mainly occur near the discontinuity point. The dominancy of the back-thrust is visible in the deformation style which is dipped toward the displacing side wall. However, in the previous cases with only side wall displacing, deformation is dipped in the same direction of the displacement. This variation in the dominancy of the back-thrusts or fore-thrusts and the shape of the deformation can be used as valuable insights for assigning the correct

boundary condition. Additionally, discrepancies between previous studies can be clarified by examining the differences in their assigned boundary conditions.

A parameter that has been considered only to a minor degree in modeling of back-thrust formation so far is the pre-consolidation pressure (i.e. material strength). Its effect on back-thrust formation apparently predominates other parameters such as basal friction, material properties, or loading, which therefore stresses the need for a better understanding of its significance in geologic settings.

**5 Conclusion**

This study systematically investigates the results of 42 numerical simulations to changes in material properties, taper angle, friction coefficient, and boundary conditions, with a specific focus on back-thrust development. The results highlight the significance of material strength (as the most important factor) and emphasize the hindering impact of increased friction along the décollement (as the second factor). The taper angles also play a role, with reduced décollement dip angles facilitating back-thrusting while surface dip angles tend to impede this behaviour. The research categorizes scenarios indicating the presence or absence of back-thrusts based on the mentioned sensitivity parameters. Occurrence of back-thrusting depends on the contrast between the taper strength and décollement strength. If sliding of taper along the décollement is easy, but the material of the taper is strong and pushes back, back-thrusting is likely to occur. However, if sliding of the taper along décollement requires more work than shearing the taper the décollement detaches to a new décollement resulting in fore-thrust formation. A constant displacement rate along the décollement as an additional boundary condition increases this effect and therefore yields more pronounced back-thrust development. Key findings of our research not only confirm previous studies, with special emphasis on the influence of material strength and basal friction in general. Our study also provides new evidence and practical rule-of-thumb guidelines to explain and understand the formation of back-thrusts in the frontal part of fold-and-thrust-belt systems and point at additional studies, which address the effect of pore pressure, diagenesis and sedimentation on the tectonic style of compressive settings.

**Competing interests**

The contact author has declared that none of the authors has any competing interests.

**Acknowledgement**

This work was funded by the Bavarian State Ministry of Science and the Arts through the framework of the Geothermal-Alliance Bavaria (GAB). The authors would like to the Rockfield team (especially William Ferguson) for providing the Elfen software package and technical discussions. The authors sincerely thank the editor (Prof. Susanne Buiter) and reviewers (Dr. David Hindle & Prof. Pauline Souloumiac) for taking the time to review our manuscript, particularly Dr. David Hindle for his useful help regarding the sensitivity analysis, and providing constructive feedback to improve our manuscript.

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
