# Peer review of "Numerical Investigation of Parameters Influencing Back-Thrust Development in Outer Wedge Fronts of Fold-and-Thrust-Belt Systems"

_EGUsphere, 2024_

## Referee Comment (RC1)

Comments on the manuscript "Numerical Investigation of Parameters Influencing Back-Thrust Development in Outer Wedge Fronts of Fold-and-Thrust-Belt Systems. " by Saeed Mahmoodpour et al.

**General comments**

The article presents a numerical approach that aims to identify the parameters and boundary conditions for which the development of back-thrusts predominates in fold-and-thrust belt systems. The authors performed simulations for 36 different prototypes. The prototype comprises six layers, including three for the basement and one for surface deposition. The parameters tested are the strength of the material, the friction value on the basal décollement, the dip of the basal décollement and the displacements imposed for the calculation. The authors show that back thrusts occur when the bulk material is resistant and friction on the basal décollement is low.

Globally, this article is rather well written but English is not my first language. Therefore, I will not comment on the language used.
The origin and mechanical cause of the vergence change in the fold-and-thrust belts remain poorly studied. Consequently, the question of this study is very interesting. Furthermore, the use of numerical modelling enables the examination of numerous models, thereby facilitating the elucidation of the pertinent question. Some studies have been carried out in sandboxes. Gutscher et al (2001) highlighted the development of BT. However, they show a single sandbox experiment with the backward vergence. The findings of the backward vergence are also outlined in the paper by Zhou and Zhou (2021) and they indicate that this vergence alteration is attributable to the lateral/basal shear stress ratio. Nevertheless, in the context of sandbox utilization, it remains uncertain whether this constitutes an experimental bias. This issue is not present in the simulations presented in this article, as they are two-dimensional numerical simulations.

I will first give my main remarks and then comment by section.

My main concerns are that:
(1) *Definition of Back-Thrusts:* while the figures presented is readily comprehensible, discerning the distinctions between each case tested is challenging. However, what do you really mean by 'Back-Thrust (BT)'? Is it a vergence towards the rear of the wedge with a succession of faults that chronologically develop towards the back wall – as landward sequence? Or is it a main ramp associated with several BTs, as can be seen in sandboxes? These BTs are concomitant with ramps or they are out of sequence? I don't think it's clear from the figures in the article where the BTs you want to show are located. In general, it is advisable to indicate the location of faults by placing arrows on each figure. Moreover, it would be beneficial to present the results in a more concise manner, or to provide guidance to the reader regarding the specific details that should be observed in order to facilitate comparison between each case.
In the introduction, the authors cite the works of Cubas et al (2016) and MacKay (1995) with regard to the orientation of the stress direction relative to the décollement. This aspect was previously demonstrated in studies conducted prior to the publication of these articles. Cubas et al (2016) demonstrated that a significant friction contrast between the décollement and the bulk material of the wedge is necessary for the formation of landward sequence. This is also demonstrated in this present study. But Cubas et al (2016) add that the wedge must be to get very close to the upper limit of the Dahlen envelope (1984), implied a steep surface slope. In this case, the major fault that develops in the wedge is a BT, with an associated fore-thrust. If the slope is really too steep, gravity slides will be observed at the rear of the wedge.
I think that in the present study it would be judicious to show a result with a strong sedimentation of surface in order to increase the slope of surface of the wedge, with a strong material strength of the bulk and a very weak friction at the base of the wedge. Perhaps it has already been calculated among the 36 cases, but if so, I haven't found it.

(2) *Mechanical criterion:* in addition, to make it easier to read, it would be advisable to explain the mechanical behavior law used a little more explicitly: the SR3 yield surface. Why p and q do not appear in the equation 4? What exactly does represent the parameter β in the equation 4? Usually, β is used for the basal dip in the Wedge Theory. Why is β so strong (Table 2)?

What do you mean when you mention the strength of material? Is it in terms of cohesion or just pre-consolidation?

Finally, in table 2, are we to understand that there are specific parameters for layers 4 and 5 and that all the other layers have the same parameters?

(3) *Plastic strain criterion:* what criteria did you use to categorize the results? You mention a critical value of 20 (line 125). What does this mean? You also mention a mesh size with an associated plastic deformation value (lines 127-130). Could you explain this part?

Finally, how is the porosity calculated?

(4) *Input parameters:* you need to check the parameters. I think there's some confusion between β and α. The parameter α is commonly used for the surface dip and β for the basal décollement dip. It would be simpler to use this nomenclature to make it easier to read. On the figure 1, α is noted at the base of the layer 4… and as I pointed out above, β is used for a friction parameter. It's confusing.

Concerning boundary conditions in displacement, I understand that for most models you require a pure displacement of the back wall. And for some (*) you also require the base to be able to move. What conclusions do you draw about the development of back-trusts linked to this specific displacement condition? How do you explain this difference of boundary conditions in nature?

(5) *Initial prototype and activation of the décollement:* do the top of layer 3 and the base of layer 2 have the same dip? Why are there three layers in the basement with different rheological parameters (according to table 2)?

According to the diagram on the Figure 1, the décollement can only be activated up to the black cross. Would it be possible to indicate with a line on each result figure which part of the décollement is actually activated? In most of the results presented, the décollement seems to have reached the plastic limit right up to the front of the wedge at 2 Ma. Then the plasticity in this basal layer seems to change over time. What we don't know is whether décollement is activated beyond the root of the fault formed, which is possible if the wedge is in a critical state in the Dahlen sense.

(6) *References:* The authors provide a comprehensive introduction, which encompasses both the numerical modelling articles and the results of the analogue sandbox modelling. However, it seems pertinent to highlight the contributions based on the Limit Analysis Theory applied to fold-and-thrust belts. In particular, Mary et al. (2013) have demonstrated that the location of faults and their lifetimes are based on deterministic chaos. The work by Adwan et al (2024), which has recently been published in Solid Earth EGU, will enable the authors to compare their results in terms of stress values. Finally, Robert et al (2019) have investigated the impact of syn-tectonic sedimentation on the stresses in a ramp propagation fold. These works can be used as a basis for discussion or can be cited in the introduction.

In the following sections, I will give comments and suggestions to particular points in the text. Numbers are line numbers. I hope my comments are useful and constructive.
Best regards,
Pauline Souloumiac.

**Section comment**

**Section 2.1:**
127 - 130: Please, clarify these criteria.

**Section 2.2:**
141: the values of the parameters written in this way are not clear. present them in the form of a list.
In this section, you should quote Figure 1 for a better understanding of the geometry.

**Section 2.3:**
204: I don't understand the meaning of the parameter $n_{sr3}$.

**Section 2.4:**
231: This word "coefficient" is missing.

**Section 3.1:**
267: What criteria do you use to describe an effective thrust? This ties in with my general comments on mechanical criterion.

**Section 3.3:**
348 - 355: it is challenging for the reader to ascertain the outcomes, as they are not explicitly in the primary text.

**Section 4.1:**
374-379: This part is not clear to me. I don't understand why the layer deposited during the folding sequence has to be pre-consolidated. Is this specifically due to the choice of SR3 yield surface criterion?

391: If the ramp develops at the rear of the structure, I think that's explained by the Coulomb critical wedge theory: the wedge is unstable at the start of the folding sequence.

**Section 4.2:**
422: add unit "degrees" for the dip angle.

**Section 4.4:**
465: replace "taper strength" by "bulk strength".

**Table 1:**
Replace "β in Eq.2" by "β in Eq.4".

**Table 2:**
Check the name of the dip of the décollement

**Figure 4:**
Do you consider that the pop-up at the back of the wedge represents the BT?

**Figure 6:**
why there are two incipient pop-ups at 2Ma and a major ramp rooted from the right basal corner at 4Ma?

**Figure 7:**
d) How can you explain the high stress values at the surface of the model?

**Figure 8:**
Why does the décollement appear to be fully activated at 2 Ma? Whereas this is no longer the case at 4 Ma. Or maybe the layer underneath was completely plastic and then it wasn't?

**Suggested additional references:**

Mary, B. C. L., Maillot, B., & Leroy, Y. M. (2013). Deterministic chaos in frictional wedges revealed by convergence analysis. *International Journal for Numerical and Analytical Methods in Geomechanics*, *37*(17), 3036-3051.

Adwan, A., Maillot, B., Souloumiac, P., Barnes, C., Nussbaum, C., Rahn, M., & Van Stiphout, T. (2024). Understanding the stress field at the lateral termination of a thrust fold using generic geomechanical models and clustering methods. *Solid Earth*, *15*(12), 1445-1463.

Robert, R., Souloumiac, P., Robion, P., & David, C. (2019). Numerical simulation of deformation band occurrence and the associated stress field during the growth of a fault-propagation fold. *Geosciences*, *9*(6), 257.

---

## Referee Comment (RC2)

[referee-annotated manuscript omitted]

---

## Author Comment (AC1)

**Reviewer #1**

Comments on the manuscript "Numerical Investigation of Parameters Influencing Back Thrust Development in Outer Wedge Fronts of Fold-and-Thrust-Belt Systems." by Saeed Mahmoodpour et al.

**General comments**

The article presents a numerical approach that aims to identify the parameters and boundary conditions for which the development of back-thrusts predominates in fold-and-thrust belt systems. The authors performed simulations for 36 different prototypes. The prototype comprises six layers, including three for the basement and one for surface deposition. The parameters tested are the strength of the material, the friction value on the basal décollement, the dip of the basal décollement and the displacements imposed for the calculation. The authors show that back thrusts occur when the bulk material is resistant and friction on the basal décollement is low. Globally, this article is rather well written but English is not my first language. Therefore, I will not comment on the language used. The origin and mechanical cause of the vergence change in the fold-and-thrust belts remain poorly studied. Consequently, the question of this study is very interesting. Furthermore, the use of numerical modelling enables the examination of numerous models, thereby facilitating the elucidation of the pertinent question. Some studies have been carried out in sandboxes. Gutscher et al (2001) highlighted the development of BT. However, they show a single sandbox experiment with the backward vergence. The findings of the backward vergence are also outlined in the paper by Zhou and Zhou (2021) and they indicate that this vergence alteration is attributable to the lateral/basal shear stress ratio. Nevertheless, in the context of sandbox utilization, it remains uncertain whether this constitutes an experimental bias. This issue is not present in the simulations presented in this article, as they are two-dimensional numerical simulations. I will first give my main remarks and then comment by section.

**Response:** Thank you very much for your kind letter and valuable comments. Your feedback is highly instructive and has helped improve the manuscript. Based on your suggestions and comments, we have made the necessary revisions and hope that the current version is more readable. Below are our responses to each comment.

**My main concerns are that:**

(1) Definition of Back-Thrusts: while the figures presented is readily comprehensible, discerning the distinctions between each case tested is challenging. However, what do you really mean by 'Back-Thrust (BT)'? Is it a vergence towards the rear of the wedge with a succession of faults that chronologically develop towards the back wall – as landward sequence? Or is it a main ramp associated with several BTs, as can be seen in sandboxes? These BTs are concomitant with ramps or they are out of sequence? I don't think it's clear from the figures in the article where the BTs you want to show are located. In general, it is advisable to indicate the location of faults by placing arrows on each figure. Moreover, it would be beneficial to present the results in a more concise manner, or to provide guidance to the reader regarding the specific details that should be observed in order to facilitate comparison between each case. In the introduction, the authors cite the works of Cubas et al (2016) and MacKay (1995) with regard to the orientation of the stress direction relative to the décollement. This aspect was previously demonstrated in studies conducted prior to the publication of these articles. Cubas et al (2016) demonstrated that a significant friction contrast between the décollement and the bulk material of the wedge is

necessary for the formation of landward sequence. This is also demonstrated in this present study. But Cubas et al (2016) add that the wedge must be to get very close to the upper limit of the Dahlen envelope (1984), implied a steep surface slope. In this case, the major fault that develops in the wedge is a BT, with an associated fore thrust. If the slope is really too steep, gravity slides will be observed at the rear of the wedge. I think that in the present study it would be judicious to show a result with a strong sedimentation of surface in order to increase the slope of surface of the wedge, with a strong material strength of the bulk and a very weak friction at the base of the wedge. Perhaps it has already been calculated among the 36 cases, but if so, I haven't found it.

**Response:** The authors appreciate the concerns raised by the reviewer. We assumed the first definition provided by the reviewer for the back-thrust: a back-thrust is considered a vergence towards the rear of the wedge, with a succession of faults that chronologically develop toward the back wall—forming a landward sequence. Additionally, the input parameters for each figure have been included in the captions to facilitate easier case follow-up.

Thank you for your suggestion to insert arrows to indicate the developed faults. However, we chose not to implement this for all cases as it would make the figures unclear, and some features—such as the exact values of the effective plastic strain or stress—would be obscured by the arrows, given that the fault-developed areas are highly deformed. In Figure 3, which illustrates the deformation development in Case #1, we have placed arrows outside the plot to indicate the fore-thrust and back-thrust as examples.

Changing the surface slope and adding extra sedimentation would increase the number of required simulations. Also, it is difficult to determine the exact boundary between fore-thrust and back-thrust development due to the highly nonlinear nature of the system and cross-correlations between certain parameters. Therefore, this study is restricted to discussing the impact of these parameters on back-thrust development rather than identifying the precise boundaries between fore-thrust and back-thrust development conditions. Throughout the text, the possible impacts of the surface slope are discussed. However, to gain better insight, we ran Case #27 ( $\beta = 2^{\circ}$ ,  $\alpha = -2^{\circ}$ ,  $\mu$ decol. = 0, and pc = 10 M) with a new surface slope of  $\alpha = -6^{\circ}$ , as suggested by the reviewer. We observed a weak back-thrust at 7 Ma (back-thrust with plastic strain lower than 1), same as for a surface slope of  $\alpha = -2^{\circ}$  in which the developed back-thrust was weak. This new simulation shows that even steeper surface slope does not make changes to our analysis. It should be noted that in this new system, the deposition baseline is placed at a distance of 14 km from the bottom of the system, whereas it was 11 km in Case #27.

**Figure r1: Effective plastic strain for case with $\beta = 2^\circ$ , $\alpha = -6^\circ$ , $\mu decol. = 0$ , and pc = 10 MPa at 7 Ma**

(2) Mechanical criterion: in addition, to make it easier to read, it would be advisable to explain the mechanical behavior law used a little more explicitly: the SR3 yield surface. Why p and q do not appear in the equation 4? What exactly does represent the parameter  $\beta$  in the equation 4? Usually,  $\beta$  is used for the basal dip in the Wedge Theory. Why is  $\beta$  so strong (Table 2)? What do you mean when you mention the strength of material? Is it in terms of cohesion or just pre consolidation? Finally, in table 2, are we to understand that there are specific parameters for layers 4 and 5 and that all the other layers have the same parameters?

**Response:** Equation 4 shows the slope of the critical state line, which is a material property that is independent of the stress state. Based on this, p and q do not appear in the equation 4.

In the revised version,  $\xi$  is used instead of  $\beta$  to show the friction parameter.  $\xi$  is similar to the friction angle based on the degree (°) values and a controlling parameter on the shape of the yield surface. Eq. A1 shows its contribution on the yield surface equation. The following example may show its impact. Following parameters are considered to plot yield surfaces based on the different  $\xi$  values.

| Parameter       | Definition                                          | Value (unit)   |
|-----------------|-----------------------------------------------------|----------------|
| $\varphi_{ref}$ | Reference porosity                                  | 0.3 (-)        |
| pt              | Reference tensile intercept                         | 0.5 (MPa)      |
| рс              | Reference compressional intercept                   | -50 (MPa)      |
| ξ               | Material constant P'-Q Plane                        | 50, 55, 60 (°) |
| Ψ               | Material constant deviatoric plane                  | 50 (°)         |
| $\xi_0^{\pi}$   | Material constant deviatoric plane                  | 0.6 (-)        |
| $\xi_1^{\pi}$   | Material constant deviatoric plane                  | 0.001 (1/MPa)  |
| N               | Material constant deviatoric plane                  | 0.25 (-)       |
| n               | Material constant P'-Q Plane                        | 1.4 (-)        |
| λ               | Plastic consolidation trend                         | 0.12 (-)       |
| Ro              | Initial yield surface ratio to failure/peak surface | 0.36(-)        |

Table r1: Input parameters to obtain yield surface shape for different friction parameter ( $\xi$ ) values

Figure r2: Impact of the friction parameter of  $\xi$  on the yield surface

Throughout the manuscript, it is assumed that a material with higher pre-consolidation pressure is stronger than a material with lower pre-consolidation pressure. In the supplementary file, some of the primary equations related to SR3 are provided. However, for a more detailed explanation and mathematical background, the cited references are recommended.

**Regarding the parameters provided in Table 2: Yes, that is correct.**

(3) Plastic strain criterion: what criteria did you use to categorize the results? You mention a critical value of 20 (line 125). What does this mean? You also mention a mesh size with an associated plastic deformation value (lines 127-130). Could you explain this part? Finally, how is the porosity calculated?

**Response:** The value of 20 for distortion area error is a controlling parameter used to activate the remeshing procedure. To reduce computational time and improve the chances of convergence, a remeshing scheme is implemented throughout the simulation. This scheme allows for the use of a larger mesh size in less affected areas, while a smaller mesh size is applied in highly affected areas during deformation. In the next section, the relationship between mesh size and plastic strain (which indicates the intensity of deformation) is provided:

- Mesh size = 400m for Plastic strain = 0
- Mesh size = 400m for Plastic strain = 0.1
- Mesh size = 300m for Plastic strain = 0.2
- Mesh size = 200m for Plastic strain > 0.5

Based on this, for areas that are either unaffected or only slightly affected by deformation, a mesh size of 400 m is used. In contrast, for highly affected areas where the plastic strain exceeds 0.5, a smaller mesh size of 200 m is implemented.

Additionally, as mentioned in line 240 of the old version, the maximum plastic strain in the developed fore-thrust and back-thrust is used to categorize weak and strong thrusts.

Specifically, a maximum effective plastic strain of 0.1 and 0.5 is used as the boundary between N, Y-W, and Y-S at early times (2 Ma), while 0.5 and 1 is used at late times (7 Ma).

During the simulation, based on the material's elastic-plastic properties, the volumetric strain for both the elastic and plastic components is calculated, and porosity values are updated accordingly.

(4) Input parameters: you need to check the parameters. I think there's some confusion between  $\beta$  and  $\alpha$ . The parameter  $\alpha$  is commonly used for the surface dip and  $\beta$  for the basal décollement dip. It would be simpler to use this nomenclature to make it easier to read. On the figure 1,  $\alpha$  is noted at the base of the layer 4... and as I pointed out above,  $\beta$  is used for a friction parameter. It's confusing. Concerning boundary conditions in displacement, I understand that for most models you require a pure displacement of the back wall. And for some (\*) you also require the base to be able to move. What conclusions do you draw about the development of back-trusts linked to this specific displacement condition? How do you explain this difference of boundary conditions in nature?

**Response:** The authors appreciate the reviewer's suggestions. In the revised version,  $\alpha$  and  $\beta$  are used as recommended.

Regarding the boundary conditions, it should be noted that both types have been previously used in numerical simulations of back-thrust development. Here, both are examined to highlight the potential differences resulting from the different boundary conditions. The lateral boundary conditions are derived from plate boundary convergence in real cases. A combination of lateral and bottom boundary displacements can be applied in systems where the material is strong enough for the entire block placed over the décollement to move as a solid unit.

(5) Initial prototype and activation of the décollement: do the top of layer 3 and the base of layer 2 have the same dip? Why are there three layers in the basement with different rheological parameters (according to table 2)? According to the diagram on the Figure 1, the décollement can only be activated up to the black cross. Would it be possible to indicate with a line on each result figure which part of the décollement is actually activated? In most of the results presented, the décollement seems to have reached the plastic limit right up to the front of the wedge at 2 Ma. Then the plasticity in this basal layer seems to change over time. What we don't know is whether décollement is activated beyond the root of the fault formed, which is possible if the wedge is in a critical state in the Dahlen sense.

**Response:** This study is a preliminary investigation into the development of back-thrusts in the Molasse Basin, southern Germany. In this configuration, there are six layers; however, at this stage, we aimed to keep the model generic. The primary focus of this manuscript is on the top layers, where back-thrusting occurs. Therefore, we believe that the number of base layers does not significantly affect the results. Instead, the friction coefficient and the dip angle of the décollement are the key influencing factors.

Regarding the location of the discontinuity point along the décollement, it has now been indicated in all figures in the revised version. For the development of plastic strain beyond the discontinuity point, we have used Figure 8 as an example, as this issue is raised in subsequent comments. Based on the mentioned figure, the layer beneath the décollement exhibits plastic strain due to stress development within the system. At 2 Ma, the plastic strain range is 0–1, making it visible. However, at 4 Ma, the data range is 0–2.7, causing the small plastic strain to be

overshadowed. In the next figure, we have adjusted the data range at 4 Ma to 0–1, making the plastic strain visible at this time as well.

Figure r3: Effective plastic strain for case#7 at 4 Ma

(6) References: The authors provide a comprehensive introduction, which encompasses both the numerical modelling articles and the results of the analogue sandbox modelling. However, it seems pertinent to highlight the contributions based on the Limit Analysis Theory applied to fold-and-thrust belts. In particular, Mary et al. (2013) have demonstrated that the location of faults and their lifetimes are based on deterministic chaos. The work by Adwan et al (2024), which has recently been published in Solid Earth EGU, will enable the authors to compare their results in terms of stress values. Finally, Robert et al (2019) have investigated the impact of syn-tectonic sedimentation on the stresses in a ramp propagation fold. These works can be used as a basis for discussion or can be cited in the introduction. In the following sections, I will give comments and suggestions to particular points in the text. Numbers are line numbers. I hope my comments are useful and constructive.

**Response:** Thank you for introducing these references. The following section has been included in the revised version:**

Limit Analysis Theory has also been applied in the literature to study fold-and-thrust belt systems. In this regard, Mary et al. (2013) demonstrated that the location of faults and their lifetimes are governed by deterministic chaos. Robert et al. (2019) used this approach to investigate the impact of syn-tectonic sedimentation on stresses in a ramp propagation fold. These stress values are essential for examining fracture development, their orientation, and the resulting fluid flow patterns in basin analysis. Adwan et al. (2024) applied the limit analysis approach to study stress distribution at the lateral termination of a thrust fold system. The fast run time of simulations using this approach enabled the authors to conduct a high number of simulations to analyse the effects of basement and fault friction angles on the failure pattern.

**Best regards,**

Pauline Souloumiac.

**Section comment**

**Section 2.1: 127 - 130: Please, clarify these criteria.**

**Response:** To reduce computational time and improve the likelihood of convergence, a remeshing scheme is implemented throughout the simulation. This scheme allows for the use of a larger mesh size in less affected areas, while a smaller mesh size is applied in highly affected regions during deformation. In the next section, we provide an analysis of the relationship between mesh size and plastic strain, which indicates the intensity of deformation:

- Mesh size = 400m for Plastic strain = 0
- Mesh size = 400m for Plastic strain = 0.1
- Mesh size = 300m for Plastic strain = 0.2
- Mesh size = 200m for Plastic strain > 0.5

Based on this, for areas that are unaffected or only slightly affected by deformation, a mesh size of 400m is used. In contrast, for highly affected areas where the plastic strain exceeds 0.5, a smaller mesh size of 200m is implemented.

**Section 2.2: 141: the values of the parameters written in this way are not clear. present them in the form of a list. In this section, you should quote Figure 1 for a better understanding of the geometry.**

**Response**: Thanks for this suggestion. It is considered in the revised version.

**Section 2.3: 204: I don't understand the meaning of the parameter nsr3.**

**Response**: It is a material specific parameter (material constant) which is used to obtain the slope of the SR3 critical state line. Considering the mentioned reference in the manuscript (Gao et al. 2018), in a system with friction parameter ( $\xi$ ) of 60° and material constant (nsr3) of 1.3, the slop of the SR3 critical state line would be:

$$\eta_{cs} = \tan 60^{\circ} \left[ (1.3 + 1)^{-\frac{1}{1.3}} \right] \approx 0.91$$

**Section 2.4: 231: This word "coefficient" is missing.**

**Response**: Thanks for finding that.

**Section 3.1: 267: What criteria do you use to describe an effective thrust? This ties in with my general comments on mechanical criterion.**

**Response**: The maximum effective plastic strain of 0.1 and 0.5 is used for boundary between nothrust, weak-thrust and strong-thrust respectively at early times (2 Ma), while 0.5 and 1 is used at late times (7 Ma).

**Section 3.3: 348 - 355: it is challenging for the reader to ascertain the outcomes, as they are not explicitly in the primary text.**

**Response**: We agree with the reviewer; however, including all figures in the main text is not feasible. Therefore, the authors have decided to place them in the appendix. We hope that interested readers will be able to refer to them in that section.

**Section 4.1: 374-379: This part is not clear to me. I don't understand why the layer deposited during the folding sequence has to be pre-consolidated. Is this specifically due to the choice of SR3 yield surface criterion? 391: If the ramp develops at the rear of the structure, I think that's explained by the Coulomb critical wedge theory: the wedge is unstable at the start of the folding sequence.**

**Response:** It should be noted that deposition does not occur as a continuous process throughout the simulation but rather in discrete time steps, which span thousands of years. Considering this phenomenon within the numerical simulation tool, it is reasonable to assume that the deposited materials are already pre-consolidated. This assumption helps mitigate divergence issues during the simulation. Additionally, we used pre-consolidation pressure as a parameter to illustrate differences in material strength; however, it does not strictly correspond to the previously experienced stress path of the material.

**Section 4.2: 422: add unit "degrees" for the dip angle.**

**Response**: Thanks for finding this missing point.

**Section 4.4: 465: replace "taper strength" by "bulk strength".**

**Response: Done.**

**Table 1: Replace " $\beta$  in Eq.2" by " $\beta$  in Eq.4".**

**Response**: The authors may have misunderstood the point, possibly due to a typographical error in the equation numbering.

**Table 2: Check the name of the dip of the décollement**

**Response**:  $\beta$  is used in the revised version as suggested.

**Figure 4: Do you consider that the pop-up at the back of the wedge represents the BT? Equal strength between fore-thrust and back-thrust**

**Response:** The authors aimed to closely approximate the natural boundary conditions and material properties. The simulation was conducted using a forward geo-mechanical scheme governed by physical laws. In some cases, such a structure may emerge due to specific conditions.

**Figure 6: why there are two incipient pop-ups at 2Ma and a major ramp rooted from the right basal corner at 4Ma?**

**Response:** Thank you for raising this issue. If I understand the question correctly, the reviewer's concern is about the pop-up structure visible at 2 Ma but no longer apparent at 4 Ma. This occurs due to the different plastic strain ranges used in Figure 6a and Figure 6b. If the data range in Figure 6b is adjusted to 0–1 (the same as in Figure 6a), the mentioned pop-up structure would become visible, as shown in the following figure.

Figure r4: Effective plastic strain at 4 Ma for case #11

Another concern is the initiation point of the fore-thrust, which originates from the right corner instead of the discontinuity point on the décollement. This can be explained by the high friction coefficient over the décollement in this case ( $\mu$ \_deco

---

## Author Comment (AC2)

**Reviewer #1**

Comments on the manuscript "Numerical Investigation of Parameters Influencing Back Thrust Development in Outer Wedge Fronts of Fold-and-Thrust-Belt Systems. " by Saeed Mahmoodpour et al.

**General comments**

The article presents a numerical approach that aims to identify the parameters and boundary conditions for which the development of back-thrusts predominates in fold-and-thrust belt systems. The authors performed simulations for 36 different prototypes. The prototype comprises six layers, including three for the basement and one for surface deposition. The parameters tested are the strength of the material, the friction value on the basal décollement, the dip of the basal décollement and the displacements imposed for the calculation. The authors show that back thrusts occur when the bulk material is resistant and friction on the basal décollement is low. Globally, this article is rather well written but English is not my first language. Therefore, I will not comment on the language used. The origin and mechanical cause of the vergence change in the fold-and-thrust belts remain poorly studied. Consequently, the question of this study is very interesting. Furthermore, the use of numerical modelling enables the examination of numerous models, thereby facilitating the elucidation of the pertinent question. Some studies have been carried out in sandboxes. Gutscher et al (2001) highlighted the development of BT. However, they show a single sandbox experiment with the backward vergence. The findings of the backward vergence are also outlined in the paper by Zhou and Zhou (2021) and they indicate that this vergence alteration is attributable to the lateral/basal shear stress ratio. Nevertheless, in the context of sandbox utilization, it remains uncertain whether this constitutes an experimental bias. This issue is not present in the simulations presented in this article, as they are two-dimensional numerical simulations. I will first give my main remarks and then comment by section.

**Response:** Thank you very much for your kind letter and valuable comments. Your feedback is highly instructive and has helped improve the manuscript. Based on your suggestions and comments, we have made the necessary revisions and hope that the current version is more readable. Below are our responses to each comment.

**My main concerns are that:**

**(1)** Definition of Back-Thrusts: while the figures presented is readily comprehensible, discerning the distinctions between each case tested is challenging. However, what do you really mean by 'Back-Thrust (BT)'? Is it a vergence towards the rear of the wedge with a succession of faults that chronologically develop towards the back wall – as landward sequence? Or is it a main ramp associated with several BTs, as can be seen in sandboxes? These BTs are concomitant with ramps or they are out of sequence? I don't think it's clear from the figures in the article where the BTs you want to show are located. In general, it is advisable to indicate the location of faults by placing arrows on each figure. Moreover, it would be beneficial to present the results in a more concise manner, or to provide guidance to the reader regarding the specific details that should be observed in order to facilitate comparison between each case. In the introduction, the authors cite the works of Cubas et al (2016) and MacKay (1995) with regard to the orientation of the stress direction relative to the décollement. This aspect was previously demonstrated in studies conducted prior to the publication of these articles. Cubas et al (2016) demonstrated that a significant friction contrast between the décollement and the bulk material of the wedge is

necessary for the formation of landward sequence. This is also demonstrated in this present study. But Cubas et al (2016) add that the wedge must be to get very close to the upper limit of the Dahlen envelope (1984), implied a steep surface slope. In this case, the major fault that develops in the wedge is a BT, with an associated fore thrust. If the slope is really too steep, gravity slides will be observed at the rear of the wedge. I think that in the present study it would be judicious to show a result with a strong sedimentation of surface in order to increase the slope of surface of the wedge, with a strong material strength of the bulk and a very weak friction at the base of the wedge. Perhaps it has already been calculated among the 36 cases, but if so, I haven't found it.

**Response:** The authors appreciate the concerns raised by the reviewer. We assumed the first definition provided by the reviewer for the back-thrust: a back-thrust is considered a vergence towards the rear of the wedge, with a succession of faults that chronologically develop toward the back wall—forming a landward sequence. Additionally, the input parameters for each figure have been included in the captions to facilitate easier case follow-up.

Thank you for your suggestion to insert arrows to indicate the developed faults. However, we chose not to implement this for all cases as it would make the figures unclear, and some features—such as the exact values of the effective plastic strain or stress—would be obscured by the arrows, given that the fault-developed areas are highly deformed. In Figure 3, which illustrates the deformation development in Case #1, we have placed arrows outside the plot to indicate the fore-thrust and back-thrust as examples.

Changing the surface slope and adding extra sedimentation would increase the number of required simulations. Also, it is difficult to determine the exact boundary between fore-thrust and back-thrust development due to the highly nonlinear nature of the system and cross-correlations between certain parameters. Therefore, this study is restricted to discussing the impact of these parameters on back-thrust development rather than identifying the precise boundaries between fore-thrust and back-thrust development conditions. Throughout the text, the possible impacts of the surface slope are discussed. However, to gain better insight, we ran Case #27 ($\beta = 2°, \alpha = -2°, \mu\text{decol.} = 0, \text{and pc} = 10 \text{ M}$) with a new surface slope of $\alpha = -6°$, as suggested by the reviewer. We observed a weak back-thrust at 7 Ma (back-thrust with plastic strain lower than 1), same as for a surface slope of $\alpha = -2°$ in which the developed back-thrust was weak. This new simulation shows that even steeper surface slope does not make changes to our analysis. It should be noted that in this new system, the deposition baseline is placed at a distance of 14 km from the bottom of the system, whereas it was 11 km in Case #27.

[Figure]

[Figure]

Figure r1: Effective plastic strain for case with $\beta = 2°, \alpha = -6°, \mu\text{decol.} = 0$, and pc $= 10$ MPa at 7 Ma

(2) Mechanical criterion: in addition, to make it easier to read, it would be advisable to explain the mechanical behavior law used a little more explicitly: the SR3 yield surface. Why p and q do not appear in the equation 4? What exactly does represent the parameter β in the equation 4? Usually, β is used for the basal dip in the Wedge Theory. Why is β so strong (Table 2)? What do you mean when you mention the strength of material? Is it in terms of cohesion or just pre consolidation? Finally, in table 2, are we to understand that there are specific parameters for layers 4 and 5 and that all the other layers have the same parameters?

**Response:** Equation 4 shows the slope of the critical state line, which is a material property that is independent of the stress state. Based on this, p and q do not appear in the equation 4.

In the revised version, ξ is used instead of β to show the friction parameter. ξ is similar to the friction angle based on the degree (°) values and a controlling parameter on the shape of the yield surface. Eq. A1 shows its contribution on the yield surface equation. The following example may show its impact. Following parameters are considered to plot yield surfaces based on the different ξ values.

Table r1: Input parameters to obtain yield surface shape for different friction parameter (ξ) values

| Parameter | Definition | Value (unit) |
|---|---|---|
| $\varphi_{ref}$ | Reference porosity | 0.3 (-) |
| pt | Reference tensile intercept | 0.5 (MPa) |
| pc | Reference compressional intercept | -50 (MPa) |
| ξ | Material constant P'-Q Plane | 50, 55, 60 (°) |
| ψ | Material constant deviatoric plane | 50 (°) |
| $\xi_0^\pi$ | Material constant deviatoric plane | 0.6 (-) |
| $\xi_1^\pi$ | Material constant deviatoric plane | 0.001 (1/MPa) |
| N | Material constant deviatoric plane | 0.25 (-) |
| n | Material constant P'-Q Plane | 1.4 (-) |
| λ | Plastic consolidation trend | 0.12 (-) |
| $R_0$ | Initial yield surface ratio to failure/peak surface | 0.36 (-) |

[Figure]

Figure r2: Impact of the friction parameter of ξ on the yield surface

Throughout the manuscript, it is assumed that a material with higher pre-consolidation pressure is stronger than a material with lower pre-consolidation pressure. In the supplementary file, some of the primary equations related to SR3 are provided. However, for a more detailed explanation and mathematical background, the cited references are recommended.

Regarding the parameters provided in Table 2: Yes, that is correct.

(3) Plastic strain criterion: what criteria did you use to categorize the results? You mention a critical value of 20 (line 125). What does this mean? You also mention a mesh size with an associated plastic deformation value (lines 127-130). Could you explain this part? Finally, how is the porosity calculated?

**Response:** The value of 20 for distortion area error is a controlling parameter used to activate the remeshing procedure. To reduce computational time and improve the chances of convergence, a remeshing scheme is implemented throughout the simulation. This scheme allows for the use of a larger mesh size in less affected areas, while a smaller mesh size is applied in highly affected areas during deformation. In the next section, the relationship between mesh size and plastic strain (which indicates the intensity of deformation) is provided:

- Mesh size = 400m for Plastic strain = 0
- Mesh size = 400m for Plastic strain = 0.1
- Mesh size = 300m for Plastic strain = 0.2
- Mesh size = 200m for Plastic strain > 0.5

Based on this, for areas that are either unaffected or only slightly affected by deformation, a mesh size of 400 m is used. In contrast, for highly affected areas where the plastic strain exceeds 0.5, a smaller mesh size of 200 m is implemented.

Additionally, as mentioned in line 240 of the old version, the maximum plastic strain in the developed fore-thrust and back-thrust is used to categorize weak and strong thrusts.

Specifically, a maximum effective plastic strain of 0.1 and 0.5 is used as the boundary between N, Y-W, and Y-S at early times (2 Ma), while 0.5 and 1 is used at late times (7 Ma).

During the simulation, based on the material's elastic-plastic properties, the volumetric strain for both the elastic and plastic components is calculated, and porosity values are updated accordingly.

(4) Input parameters: you need to check the parameters. I think there's some confusion between $\beta$ and $\alpha$. The parameter $\alpha$ is commonly used for the surface dip and $\beta$ for the basal décollement dip. It would be simpler to use this nomenclature to make it easier to read. On the figure 1, $\alpha$ is noted at the base of the layer 4... and as I pointed out above, $\beta$ is used for a friction parameter. It's confusing. Concerning boundary conditions in displacement, I understand that for most models you require a pure displacement of the back wall. And for some (*) you also require the base to be able to move. What conclusions do you draw about the development of back-trusts linked to this specific displacement condition? How do you explain this difference of boundary conditions in nature?

**Response:** The authors appreciate the reviewer's suggestions. In the revised version, α and β are used as recommended.

Regarding the boundary conditions, it should be noted that both types have been previously used in numerical simulations of back-thrust development. Here, both are examined to highlight the potential differences resulting from the different boundary conditions. The lateral boundary conditions are derived from plate boundary convergence in real cases. A combination of lateral and bottom boundary displacements can be applied in systems where the material is strong enough for the entire block placed over the décollement to move as a solid unit.

(5) Initial prototype and activation of the décollement: do the top of layer 3 and the base of layer 2 have the same dip? Why are there three layers in the basement with different rheological parameters (according to table 2)? According to the diagram on the Figure 1, the décollement can only be activated up to the black cross. Would it be possible to indicate with a line on each result figure which part of the décollement is actually activated? In most of the results presented, the décollement seems to have reached the plastic limit right up to the front of the wedge at 2 Ma. Then the plasticity in this basal layer seems to change over time. What we don't know is whether décollement is activated beyond the root of the fault formed, which is possible if the wedge is in a critical state in the Dahlen sense.

**Response:** This study is a preliminary investigation into the development of back-thrusts in the Molasse Basin, southern Germany. In this configuration, there are six layers; however, at this stage, we aimed to keep the model generic. The primary focus of this manuscript is on the top layers, where back-thrusting occurs. Therefore, we believe that the number of base layers does not significantly affect the results. Instead, the friction coefficient and the dip angle of the décollement are the key influencing factors.

Regarding the location of the discontinuity point along the décollement, it has now been indicated in all figures in the revised version. For the development of plastic strain beyond the discontinuity point, we have used Figure 8 as an example, as this issue is raised in subsequent comments. Based on the mentioned figure, the layer beneath the décollement exhibits plastic strain due to stress development within the system. At 2 Ma, the plastic strain range is 0–1, making it visible. However, at 4 Ma, the data range is 0–2.7, causing the small plastic strain to be

overshadowed. In the next figure, we have adjusted the data range at 4 Ma to 0–1, making the plastic strain visible at this time as well.

[Figure]

Figure r3: Effective plastic strain for case#7 at 4 Ma

(6) References: The authors provide a comprehensive introduction, which encompasses both the numerical modelling articles and the results of the analogue sandbox modelling. However, it seems pertinent to highlight the contributions based on the Limit Analysis Theory applied to fold-and-thrust belts. In particular, Mary et al. (2013) have demonstrated that the location of faults and their lifetimes are based on deterministic chaos. The work by Adwan et al (2024), which has recently been published in Solid Earth EGU, will enable the authors to compare their results in terms of stress values. Finally, Robert et al (2019) have investigated the impact of syn-tectonic sedimentation on the stresses in a ramp propagation fold. These works can be used as a basis for discussion or can be cited in the introduction. In the following sections, I will give comments and suggestions to particular points in the text. Numbers are line numbers. I hope my comments are useful and constructive.

**Response:** Thank you for introducing these references. The following section has been included in the revised version:

Limit Analysis Theory has also been applied in the literature to study fold-and-thrust belt systems. In this regard, Mary et al. (2013) demonstrated that the location of faults and their lifetimes are governed by deterministic chaos. Robert et al. (2019) used this approach to investigate the impact of syn-tectonic sedimentation on stresses in a ramp propagation fold. These stress values are essential for examining fracture development, their orientation, and the resulting fluid flow patterns in basin analysis. Adwan et al. (2024) applied the limit analysis approach to study stress distribution at the lateral termination of a thrust fold system. The fast run time of simulations using this approach enabled the authors to conduct a high number of simulations to analyse the effects of basement and fault friction angles on the failure pattern.

Best regards,

Pauline Souloumiac.

**Section comment**

**Section 2.1: 127 - 130: Please, clarify these criteria.**

**Response:** To reduce computational time and improve the likelihood of convergence, a remeshing scheme is implemented throughout the simulation. This scheme allows for the use of a larger mesh size in less affected areas, while a smaller mesh size is applied in highly affected regions during deformation. In the next section, we provide an analysis of the relationship between mesh size and plastic strain, which indicates the intensity of deformation:

- Mesh size = 400m for Plastic strain = 0
- Mesh size = 400m for Plastic strain = 0.1
- Mesh size = 300m for Plastic strain = 0.2
- Mesh size = 200m for Plastic strain > 0.5

Based on this, for areas that are unaffected or only slightly affected by deformation, a mesh size of 400m is used. In contrast, for highly affected areas where the plastic strain exceeds 0.5, a smaller mesh size of 200m is implemented.

**Section 2.2: 141: the values of the parameters written in this way are not clear. present them in the form of a list. In this section, you should quote Figure 1 for a better understanding of the geometry.**

**Response**: Thanks for this suggestion. It is considered in the revised version.

**Section 2.3: 204: I don't understand the meaning of the parameter nsr3.**

**Response**: It is a material specific parameter (material constant) which is used to obtain the slope of the SR3 critical state line. Considering the mentioned reference in the manuscript (Gao et al. 2018), in a system with friction parameter ($\xi$) of 60° and material constant (nsr3) of 1.3, the slop of the SR3 critical state line would be:

$$\eta_{cs} = \tan 60° \, [(1.3 + 1)^{-\frac{1}{1.3}}] \approx 0.91$$

**Section 2.4: 231: This word "coefficient" is missing.**

**Response**: Thanks for finding that.

**Section 3.1: 267: What criteria do you use to describe an effective thrust? This ties in with my general comments on mechanical criterion.**

**Response**: The maximum effective plastic strain of 0.1 and 0.5 is used for boundary between no-thrust, weak-thrust and strong-thrust respectively at early times (2 Ma), while 0.5 and 1 is used at late times (7 Ma).

**Section 3.3: 348 - 355: it is challenging for the reader to ascertain the outcomes, as they are not explicitly in the primary text.**

**Response**: We agree with the reviewer; however, including all figures in the main text is not feasible. Therefore, the authors have decided to place them in the appendix. We hope that interested readers will be able to refer to them in that section.

**Section 4.1: 374-379: This part is not clear to me. I don't understand why the layer deposited during the folding sequence has to be pre-consolidated. Is this specifically due to the choice of SR3 yield surface criterion? 391: If the ramp develops at the rear of the structure, I think that's explained by the Coulomb critical wedge theory: the wedge is unstable at the start of the folding sequence.**

**Response:** It should be noted that deposition does not occur as a continuous process throughout the simulation but rather in discrete time steps, which span thousands of years. Considering this phenomenon within the numerical simulation tool, it is reasonable to assume that the deposited materials are already pre-consolidated. This assumption helps mitigate divergence issues during the simulation. Additionally, we used pre-consolidation pressure as a parameter to illustrate differences in material strength; however, it does not strictly correspond to the previously experienced stress path of the material.

**Section 4.2: 422: add unit "degrees" for the dip angle.**

**Response**: Thanks for finding this missing point.

**Section 4.4: 465: replace "taper strength" by "bulk strength".**

**Response**: Done.

**Table 1: Replace "β in Eq.2" by "β in Eq.4".**

**Response**: The authors may have misunderstood the point, possibly due to a typographical error in the equation numbering.

**Table 2: Check the name of the dip of the décollement**

**Response**: β is used in the revised version as suggested.

**Figure 4: Do you consider that the pop-up at the back of the wedge represents the BT? Equal strength between fore-thrust and back-thrust**

**Response:** The authors aimed to closely approximate the natural boundary conditions and material properties. The simulation was conducted using a forward geo-mechanical scheme governed by physical laws. In some cases, such a structure may emerge due to specific conditions.

**Figure 6: why there are two incipient pop-ups at 2Ma and a major ramp rooted from the right basal corner at 4Ma?**

**Response:** Thank you for raising this issue. If I understand the question correctly, the reviewer's concern is about the pop-up structure visible at 2 Ma but no longer apparent at 4 Ma. This occurs due to the different plastic strain ranges used in Figure 6a and Figure 6b. If the data range in Figure 6b is adjusted to 0–1 (the same as in Figure 6a), the mentioned pop-up structure would become visible, as shown in the following figure.

[Figure]

Figure r4: Effective plastic strain at 4 Ma for case #11

Another concern is the initiation point of the fore-thrust, which originates from the right corner instead of the discontinuity point on the décollement. This can be explained by the high friction coefficient over the décollement in this case (μ_decol. = 0.3). A high friction coefficient increases resistance to the displacing force, causing it to initiate from the right boundary of the system, ultimately leading to fore-thrust development in this section.

**Figure 7: d) How can you explain the high stress values at the surface of the model?**

**Response:** Thank you for carefully examining the figures. I believe the concern raised pertains to subplot (d) in the figures, where the ratio of horizontal to vertical stress is plotted. At the top surface, the vertical stress—mainly influenced by the overburden material—is relatively small. As a result, even slight changes in horizontal stress can lead to a high horizontal-to-vertical stress ratio. When considering effective stress values or vertical stress values, the top section exhibits a low stress value. For example, refer to the following figure for the effective stress of case #7* and compare it with Figure 8d.

[Figure]

Figure r5: Effective stress values (MPa) for case #7* at 7 Ma.

**Figure 8: Why does the décollement appear to be fully activated at 2 Ma? Whereas this is no longer the case at 4 Ma. Or maybe the layer underneath was completely plastic and then it wasn't?**

**Response:** Thank you for raising this concern. Based on the mentioned figure, the layer beneath the décollement exhibits plastic strain due to stress development within the system. At 2 Ma, the plastic strain range is 0–1, making this small plastic strain visible. However, at 4 Ma, the data range is 0–2.7, causing the small plastic strain to be overshadowed. In the following figure, the data range at 4 Ma has been adjusted to 0–1, making the plastic strain visible at this time as well.

[Figure]

Figure r6: Effective plastic strain for case#7 at 4 Ma

Suggested additional references:

Mary, B. C. L., Maillot, B., & Leroy, Y. M. (2013). Deterministic chaos in frictional wedges revealed by convergence analysis. International Journal for Numerical and Analytical Methods in Geomechanics, 37(17), 3036-3051.

Adwan, A., Maillot, B., Souloumiac, P., Barnes, C., Nussbaum, C., Rahn, M., & Van Stiphout, T. (2024). Understanding the stress field at the lateral termination of a thrust fold using generic geomechanical models and clustering methods. Solid Earth, 15(12), 1445-1463.

Robert, R., Souloumiac, P., Robion, P., & David, C. (2019). Numerical simulation of deformation band occurrence and the associated stress field during the growth of a fault-propagation fold. Geosciences, 9(6), 257.

===============================================================================

Reviewer #2

This article uses a highly non-linear, elastic plastic, soil model, which allows for initial porosity, compaction and plastic flow under deformation, potentially in concentrated "shear zones" in response to imposed boundary stresses. It presents a large set of numerical experiments on wedge like geometries (thrust belts) and their response to differing material and decollement properties and decollement angles. By varying experimental conditions in a semi-systematic way, a range of deformation geometries and bulk material behaviours are generated. Specifically, the study sets out to examine under what conditions "back thrusts" are generated. The final conclusions of the paper are based on a statistical analysis of the significance of the individual parameters on the propensity for back thrusting.

**Response:** Thank you for your kind letter and valuable comments. Your feedback has been highly instructive in improving the manuscript. Based on your suggestions, we have made the necessary revisions and hope that the current version is more readable. Below are our detailed responses to each comment:

Regarding this final result, it has to be said that given the experiments concern a complex system (thrust wedge) there is almost certain to be a degree of cross-correlation and covariance of parameters and their influence on back thrusting. A valid statistic ought to explore this and I am

unsure if that is the case. The details of the correlation method are not given. This is a significant issue for the paper, since it attempts to take a straightforward experimental approach to the question and thus, the correlation results are the main conclusion presented. I am not an expert in multivariate regression methods, but I am aware that this is a very large and complicated field and requires serious consideration to get right.

**Response:** The reviewer raises a valid point, and the authors agree. The primary goal of this study is to conduct a preliminary analysis of a discrete range of parameters to demonstrate their significance in back-thrust development. However, a detailed study to determine the exact relative importance of these parameters would require running a large number of simulations. Additionally, considering the cross-effects of the parameters adds further complexity and significantly increases the number of required simulations, which is beyond the scope of this study.

We rank the classification of back-thrusting from 1 (no back-thrusting) to 2 (weak back thrusting) to 3 (strong back-thrusting) and calculate the coefficient of correlation between the ranks and the varied input parameters décollement dip, décollement friction coefficient and pre-consolidation pressure (material strength) (Figure 9). The CORREL function in Excel is used to calculate the correlation coefficient between two cell ranges: one representing the back-thrusting strength weight (1, 2, 3) and the other containing the input values of each parameter, as reported in Table 2. Considering X and Y as the two data ranges, the following equation is used in Excel to perform this calculation:

$$\text{Correl(X, Y)} = \frac{\sum (x - \bar{x})(y - \bar{y})}{\sqrt{\sum (x - \bar{x})^2 (y - \bar{y})^2}}$$

Regarding the problem and the method itself, thrust wedges are, as already stated, complex systems. The plastic soild model used is a reasonable analogue for such a system, but this also ought to be discussed explicitly in the paper. The phenomenon under investigation is backthrusting, an emergent property of a complex system. Any attempt to study this problem needs to consider whether the model used is able to produce such emergent properties. There is little doubt that the critical state soil model is able to do so, but still, this should be a starting point for the paper. In this way, the initial discussion of critical wedges could be far better incorporated into the rest of the paper.

**Response:** The critical state soil model examines the elasto-plastic behavior of material while considering both deviatoric and normal stress impacts. The authors agree with the reviewer that validating the simulation outcomes with experimental data would strengthen confidence in the findings. However, finding experimental data that includes all the required input parameters and specifically examines the targeted deformation (back-thrusting) is challenging, and we were unable to find such data. Nevertheless, previous studies (such as Obradors-Prats et al. 2017) have implemented similar material models to analyse deformation styles in sandbox experiments and have observed a strong correlation between simulation results and experimental observations. Based on this, we hope that the SR3 model could be a valuable material model for back-thrust modeling. To further confirm this, future experimental studies would be beneficial.

**Obradors-Prats, J., Rouainia, M., Aplin, A, C., Crook, A, J, L. (2017). Hydromechanical modeling of stress, pore pressure and porosity evolution in fold-and-thrust belt systems. Journal of geophysical research: solid earth, 122, 9383-9403.**

The experiments themselves seem perfectly reasonable, but their presentation leaves a lot to be desired. Specifically, the experimental conditions are all placed in a single table, and then experimental results are presented by referring to the experiment's number. This is impossible to follow, since it requires endless switching back and forth between text, figures and the table to actually know what paramters are being used to get a particular result. Hence, the values of the the parameters should be visible on every figure showing model results, and ought to be incorporated into the text and discussion.

**Response:** Thank you for the suggestion. This point has been addressed in the revised version.

The scope of the experiments is a slightly different question. One thing I noticed after a while is the nature of the basal decollement angle. It is assumed to be constant (as far as I can tell). There is very good reason (due to elastic flexure) to assume any decollement at the scale of thrust belt will have a degree of elastic curvature dependent on the internal friction of the thrust wedge material. This in turn will modify the thrust wedge / critical taper response of the entire system. This was quite well dealt with by Willett and Schlunegger (2010) in a paper on the Swiss Molasse Basin/Western Alpine thrust system. You may argue that at a small enough scale, the curvature no longer has an effect. The results of these experiments are also perfectly valid as they stand. But you cannot dismiss the potential significance of the elastic bending of the lithosphere in many cases, and you should discuss this effect somewhere in the paper and how it may modify things. That is not to say you need to run more experiments (although you could if you wanted) but if you don't, then you must acknowledge the limitations of the results in the broader context of thrust wedges.

**Response:** The authors agree with the reviewer that curvature can influence stress direction. However, implementing curvature in numerical simulations is a challenging task. To approximate its potential impact, we considered complementary scenarios for case #1 and case #4, where the dip of the décollement was slightly increased at the discontinuity point (as shown in the following figure). The results were compared, and the differences were significant. Further investigation could focus on examining the likelihood of back-thrust development in a curved décollement or its approximation using combined lines with variable dips.

[Figure]

Figure r7: Schematic showing the shape of the décollement in its straight configuration (solid line) and with an increased dip (dashed line).

[Figure]

Figure r8: Impact of the varied dip of the décollement on effective plastic strain development in case #1.

[Figure]

Figure r9: Impact of the varied dip of the décollement on effective plastic strain development in case #4.

I found quite a lot of issues with the use of English in the paper. Some of these are highlighted in an annotated pdf.

**Response:** Thank you for identifying these issues. In the revised version, we have made the necessary modifications.

The reviewer suggested adding a figure to explain the following section in the text, but the authors decided to provide an explanation in the response letter.

"As reported by (Obradors-Prats et al., 2017), in a highly consolidated state, the stress path intersects the yield curve on the shear section. This intersection is crucial for modelling brittle deformation, a form of plastic deformation predictable with Cam clay-type models. Such models result in fast softening materials, eventuate to localization behavior characterized by steep displacement gradients in thin zones of strong plastic strain."

**Response:** Considering a stress path from the start point to the endpoint in the following figure, it can be observed that this path intersects the yield surface of the material with lower pre-consolidation pressure on the compaction side, while the same path intersects the yield surface of the material with higher pre-consolidation pressure on the shear side. The authors have explained this in the mentioned text:

As reported by Obradors-Prats et al. (2017), in a highly consolidated state, the likelihood of the stress path intersecting the yield curve on the shear section is higher. This intersection is crucial for modelling brittle deformation, a form of plastic deformation predictable with Cam clay-type models. Such models result in fast softening materials, eventuate to localization behavior characterized by steep displacement gradients in thin zones of strong plastic strain.

[Figure]

Figure r10: Intersection of the stress path with the yield surfaces

---

## Referee Report (RR1)

**Reflections on Statistical Interpretation in CSSM-Based Geodynamic Modeling**
Reviewer's Commentary

The revised manuscript makes several technically sound and well-supported observations regarding the influence of material strength and basal friction on the development of back-thrusts. These conclusions are consistent with critical taper theory and the results of prior experimental and numerical studies.

However, I wish to offer a reflection on a deeper methodological concern—one that goes beyond this particular paper and touches on a broader issue in the geodynamic modeling community.

The Nature of the System Under Study

The fold-and-thrust belt systems modelled here are governed by non-linear stress redistribution, critical taper dynamics, and path-dependent failure processes. The introduction of a critical state soil model, combined with large-strain elasto-plastic behavior, suggests the authors are simulating a system with characteristics of a complex system: sensitive to initial and boundary conditions, driven by conservation laws, and capable of emergent behavior such as spontaneous fault reorganization or back-thrust formation.

This implies the system is not purely deterministic, nor entirely stochastic, but something in between—a hallmark of what some would call a self-organized critical system.

The Limits of Simple Statistical Tools

In this context, the authors' use of Excel's CORREL function to evaluate pairwise correlations between ranked output categories (1 = none, 2 = weak, 3 = strong) and input parameters is problematic. While the intention to quantify parameter influence is commendable, this method:

Treats a ranked ordinal variable as though it were interval-scale,
Ignores the covariance and interdependence among inputs,
Implies univariate influence in a system whose behavior is inherently multivariate and emergent.

Such an approach risks drawing overly simplistic conclusions from what is fundamentally a high-dimensional and nonlinear process.

A Better Framework: Multivariate Thinking

Rather than correlating parameters with ranked outputs, I would encourage the use of multivariate regression analysis (linear or nonlinear), augmented with:

Variance Inflation Factor (VIF): to diagnose multicollinearity,
Standardized coefficients: to explore relative influences in standardized space,
SHAP values (Shapley Additive Explanations): to assess feature importance in a model-agnostic and interpretable way.

These tools do not "solve" the problem of emergent behavior—but they can help map the multidimensional parameter space and identify joint patterns of influence without assuming linear causality.

**Further Caveat**

The issue of scale—specifically the distinction between ordinal and interval-scale variables—is not fully resolved by the statistical methods used here. The outcome variable against which model parameters are analyzed is a dimensionless rank of back-thrust expression. These values are discrete and carry no inherent metric scale. As such, standard correlation or regression methods may not be strictly appropriate.

Proper treatment of ordinal outcomes typically requires alternative methods, such as ordinal logistic regression, which account for the ordering without assuming uniform spacing between categories. Another possibility would be to recast the outcome variable in terms of a continuous quantity—such as the amount of back-thrust displacement—if that data is available.

Nevertheless, the multivariate analysis performed here can still provide qualitative insight into patterns of parameter influence. The results should be interpreted in that light: as exploratory rather than predictive or definitive.

**On Fairness and the State of the Field**

It would be unfair to single out this paper. On the contrary, I recognize that this manuscript is ahead of many others in attempting quantitative sensitivity analysis at all. However, this step also exposes the statistical shallowness of common practice in geodynamic simulation work. Across the literature, we routinely simulate complex systems and then analyze them with tools that assume simplicity.

My intention is not to criticize this paper alone, but to suggest that we—as a community—need to reconsider how we interpret simulation results when the systems themselves are governed by emergent phenomena.

**Conclusion**

The conclusions of this study regarding material strength, basal friction, and thrust development are reasonable. But the statistical methods used to rank and relate model outcomes to input parameters do not reflect the complexity of the underlying system. In systems where emergent behavior is expected, we should be cautious in assigning causality and instead adopt tools that respect the interdependent and dynamic nature of such models.

**Appendix: Code Reference**
I have prepared a Python implementation for multivariate regression analysis, variance inflation assessment, partial correlation calculation, and SHAP-based feature attribution.

```python
import numpy as np
import pandas as pd
import statsmodels.api as sm
from sklearn.preprocessing import StandardScaler
from statsmodels.stats.outliers_influence import variance_inflation_factor
from scipy.stats import pearsonr
import shap

**=== Load Data from CSV ===**
**Replace 'data.csv' with your actual filename**
**File should have columns: X1, X2, X3, X4, Y**
df = pd.read_csv('data.csv')

**=== A) Correlation Matrix ===**
correlation_matrix = df.corr()

**=== B) Standardized Regression Coefficients ===**
scaler = StandardScaler()
X_scaled = scaler.fit_transform(df[['X1', 'X2', 'X3', 'X4']])
Y_scaled = scaler.fit_transform(df[['Y']])

X_scaled_with_const = sm.add_constant(X_scaled)
model = sm.OLS(Y_scaled, X_scaled_with_const).fit()
standardized_coefficients = model.params[1:]

**=== C) Variance Inflation Factor (VIF) ===**
vif_data = pd.DataFrame()
vif_data["Feature"] = ['X1', 'X2', 'X3', 'X4']
vif_data["VIF"] = [variance_inflation_factor(X_scaled_with_const, i+1) for i in range(4)]

**=== D) Partial Correlations ===**
partial_corrs = {}
for col in ['X1', 'X2', 'X3', 'X4']:
other_vars = [c for c in ['X1', 'X2', 'X3', 'X4'] if c != col]
model = sm.OLS(df[col], sm.add_constant(df[other_vars])).fit()
residuals_x = model.resid
model_y = sm.OLS(df['Y'], sm.add_constant(df[other_vars])).fit()
residuals_y = model_y.resid
partial_corrs[col] = pearsonr(residuals_x, residuals_y)[0]

**=== E) SHAP Values ===**
X = sm.add_constant(df[['X1', 'X2', 'X3', 'X4']])
ols_model = sm.OLS(df['Y'], X).fit()

def model_predict(X_input):
```

```
    return ols_model.predict(sm.add_constant(X_input))

explainer = shap.Explainer(model_predict, df[['X1', 'X2', 'X3', 'X4']])
shap_values = explainer(df[['X1', 'X2', 'X3', 'X4']])
shap_importance = np.abs(shap_values.values).mean(axis=0)

**=== Summary Results ===**
print("\n--- Correlation Matrix ---")
print(correlation_matrix)

print("\n--- Standardized Coefficients ---")
print(standardized_coefficients)

print("\n--- Variance Inflation Factor (VIF) ---")
print(vif_data)

print("\n--- Partial Correlations ---")
for key, value in partial_corrs.items():
print(f"{key}: {value:.3f}")

print("\n--- SHAP Feature Importance (Mean Absolute) ---")
for feature, value in zip(['X1', 'X2', 'X3', 'X4'], shap_importance):
print(f"{feature}: {value:.3f}")
```

***to use the code
Prepare a CSV file (e.g. data.csv) with columns labeled exactly as: X1, X2, X3, X4, Y.
Each row is one data point.
Save it in the same folder as the script or adjust the file path accordingly.

Other points about the paper

The original series of suggestions I made have been fairly thoroughly implemented so presentation and ease of reading are now significantly improved.

---

## Author Response (AR3)

**Editor comments**

In section 2, Methodology, could you add the mechanical equations that you are solving? That is, how is stress computed? The constitutive equations are in section 2.3, but i cannot seem to find the underlying mechanical equations. Also, could you describe your sedimentation approach and give its equation and relevant parameter values?

In addition, i would suggest to consider dividing the introduction in sub-sections as it is a fairly long section. In your revision, please also take into account the comments under "Notification to the authors from review file validation".

**Response:** The authors appreciate the concerns raised with editor.

The governing geomechanical equilibrium equations are added as:

$$\frac{\partial \sigma'_x}{\partial x} + \frac{\partial \tau_{xz}}{\partial z} = 0 \qquad\qquad\qquad\qquad\qquad \text{Eq.2}$$

$$\frac{\partial \tau_{xz}}{\partial x} + \frac{\partial \sigma'_z}{\partial z} = (\rho_b - \rho_w)g \qquad\qquad\qquad\qquad \text{Eq.3}$$

in which $\sigma'_x$, $\sigma'_z$, $\tau_{xz}$, $\rho_b$, and $\rho_w$ shows horizontal-, vertical- effective stress, shear stress, bulk-, and water- density respectively.

Regarding the sedimentation approach, the following text is added to the manuscript:

For deposition type, deposition by morphing is selected in 2 sublayers during 2 Ma. The top horizon for the deposition stage is assigned as the top surface of the Layer 0 in Figure 1. Implementing deposition by morphing would insert the new sediment in a way its upper surface is smoothly graded between the initial topographic profile (i.e., the original top of the domain) and a user-defined horizon.

The introduction section is divided into 3 subsections as:

1.1 Geological and theoretical background

1.2 Influencing factors on back-thrust development

1.3 Geomechanical forward modelling and study goal

The raised concerns during the review file validation are addressed in the revised version. The supplementary sections are numbered with S1 & S2. Also, author contributions is added to the manuscript.